# TRUST THE TYPICAL

**Debargha Ganguly**[1]* **Sreehari Sankar**[1] **Biyao Zhang**[1] **Vikash Singh**[1]
**Kanan Gupta**[2] **Harshini Kavuru**[3] **Alan Luo**[1] **Weicong Chen**[1]
**Warren Morningstar**[4] **Raghu Machiraju**[3] **Vipin Chaudhary**[1]

[1]Case Western Reserve University, [2]University of Pittsburgh
[3]The Ohio State University, [4]Google Research

## ABSTRACT

Current approaches to LLM safety fundamentally rely on a brittle cat-and-mouse game of identifying and blocking known threats via guardrails. We argue for a fresh approach: robust safety comes not from enumerating what is harmful, but from *deeply understanding what is safe*. We introduce **T**rust **T**he **T**ypical (**T3**), a framework that operationalizes this principle by treating safety as an out-of-distribution (OOD) detection problem. T3 learns the distribution of acceptable prompts in a semantic space and flags any significant deviation as a potential threat. Unlike prior methods, it requires no training on harmful examples, yet achieves state-of-the-art performance across 18 benchmarks spanning toxicity, hate speech, jailbreaking, multilingual harms, and over-refusal, reducing false positive rates by up to 40x relative to specialized safety models. A single model trained only on safe English text transfers effectively to diverse domains and over 14 languages without retraining. Finally, we demonstrate production readiness by integrating a GPU-optimized version into vLLM, enabling continuous guardrailing during token generation with less than 6% overhead even under dense evaluation intervals on large-scale workloads.

## 1 INTRODUCTION

Contemporary safety paradigms for large language models are fundamentally reactive, relying on specialized classifiers trained to detect known categories of harmful content and adversarial prompts via explicit pattern recognition (Inan et al., 2023; Deng et al., 2025; Gehman et al., 2020; Wallace et al., 2019; Carlini et al., 2021; Zou et al., 2023; Wei et al., 2023). This approach creates an inherent asymmetry, where attackers need only discover novel prompt structures that fall outside the training distribution of safety classifiers, while defenders must continuously expand their catalogs of harmful patterns – a dynamic that favors the adversary (Liu et al., 2023c). The adversarial landscape evolves continuously with new attack vectors such as multi-turn jailbreaking, role-playing exploits, and encoding-based obfuscation emerging faster than defensive systems can adapt. Consequently, even sophisticated alignment techniques like Reinforcement Learning from Human Feedback (RLHF) and Constitutional AI, while improving general alignment, cannot guarantee robustness against adversarial inputs that fall outside their training distributions (Christiano et al., 2017; Bai et al., 2022; Ouyang et al., 2022; Casper et al., 2023). These adversarial prompts succeed precisely because they share a fundamental characteristic: *they must deviate from the statistical regularities of "typical" natural language* to exploit learned vulnerabilities, a property current defenses fail to leverage systematically.

This cat-and-mouse dynamic reveals a deeper issue: current safety mechanisms can only defend against explicitly known attack patterns, and cannot anticipate and defend against novel forms of attack. We explore a paradigm shift toward proactive safety through the lens of statistical typicality. Drawing from information theory, we observe that legitimate user interactions with language models, despite their surface diversity, occupy a relatively concentrated region in the model's semantic representation space, what Cover & Thomas (1999) term the "typical set." Adversarial prompts, by design, must deviate from natural language patterns to exploit model vulnerabilities, often manifesting as atypical points in this representation space (Nalisnick et al., 2019; Morningstar et al., 2021). This

---

*This work was funded by NSF Awards 2117439 and 2112606.

reframing suggests a more efficient and robust paradigm for LLM safety. Rather than training specialized models to recognize specific harmful patterns, we can instead focus on characterizing the distribution of acceptable, in-distribution examples. Such an approach offers two key advantages. First, it obviates the need for an exhaustive and constantly updated collection of harmful examples, requiring only a specification of what constitutes safe usage. Second, by making no assumptions about the form of adversarial inputs, it provides a more principled defense against novel and unforeseen attack patterns. However, operationalizing this paradigm is challenging; one cannot directly query the true probability of a prompt under the unknown distribution of "safe and intended use." While prior works have explored statistical methods for content filtering, they often remain vulnerable to novel attacks or incur high computational overhead (Gehman et al., 2020; Xu et al., 2021).

In this paper, we introduce T3, a novel framework that fundamentally reframes LLM safety from reactive pattern-matching to proactive statistical modeling. T3 operationalizes the principle of typicality by learning the geometric structure of safe language use, then detecting deviations that characterize harmful content. Our specific contributions are:

1. We *extend* the Forte framework (Ganguly et al., 2025c) from vision to text, providing rigorous theoretical analysis of how the per-point PRDC metrics detect distributional shifts. We establish the expected values of the metrics in a much more general setting than Ganguly et al. (2025c): without any additional assumptions on the distributions in the case when samples are from the same distribution, and with mild assumptions on the density and support of the distributions in the case when they are different.

2. Across 18 benchmarks spanning toxicity, hate speech, jailbreaking, and multilingual harms, *T3 achieves state-of-the-art AUROC with a 10-40x reduction in false positive rates compared to specialized safety models* . On important benchmarks, T3-OCSVM achieves FPR@95 of 2.0% (OffensEval) and 3.5% (Davidson) versus 75.2% and 61.7% for the best baseline. This improvement directly translates to a 75% reduction in overrefusals compared to traditional methods on OR-Bench.

3. Using a single model trained only on English general-purpose safe text, T3 achieves *near-perfect transfer across specialized domains* (99.6% AUROC on code, 99.8% on HR) and *maintains consistent performance across 14+ languages* with less than 2% variance. This reduces the need for domain-specific training, multilingual data collection, or language-specific calibration.

4. We co-design T3 within vLLM to *enable continuous safety monitoring during token generation*, achieving sub-6% overhead even with aggressive 20-token evaluation intervals on 5,000-prompt workloads. By *overlapping safety computations with inference operations on the same GPU*, T3 enables immediate terminations of harmful generations without the latency penalties associated with post-processing approaches, making real-time guardrailing practical for production deployments.

## 2 RELATED WORKS

Detecting out-of-distribution (OOD) inputs is crucial for reliably deploying models, as they often yield confident but incorrect predictions on novel data, a vulnerability highlighted by adversarial perturbations (Szegedy et al., 2013) and poor calibration (Guo et al., 2017). **Supervised** OOD methods use labeled examples for explicit training (Hendrycks et al., 2019; Dhamija et al., 2018), output calibration (Liang et al., 2018; Hsu et al., 2020), ensembles (Lakshminarayanan et al., 2017), or fitting distributions to latent features (Meinke & Hein, 2020; Ganguly et al., 2025a). However, their reliance on *known* OOD examples limits effectiveness against novel threats. In contrast, **unsupervised** methods model the training data density $p(\mathbf{x})$ (Bishop, 1994), but this approach can fail in high-dimensional spaces where OOD samples receive high likelihoods (Nalisnick et al., 2019; Choi et al., 2018). Solutions like likelihood ratios (Ren et al., 2019), typicality tests (Nalisnick et al., 2019), interpretable graph-base methods (Ganguly et al., 2025a) and physics-inspired estimators (Morningstar et al., 2021) attempt to mitigate this paradox but still struggle with the curse of dimensionality.

The emergence of LLMs transformed OOD detection through the geometric property of *isotropy*, where embeddings spread uniformly in contrast to the narrow 'cones' of earlier models (Liu et al.,

Figure 1: **Geometric concentration of safe text embeddings in high-dimensional space.** The distribution of Euclidean distances from the mean for 10,000 safe embeddings (Alpaca, d=1024) empirically validates the concentration of measure phenomenon. **(a, d)** The distances closely follow a theoretical $\chi_{1024}$ distribution, confirmed by a Q-Q plot ($R^2 > 0.99$). **(b, c, f)** This results in a concentrated "typical set" where 90% of data forms an annulus ("hollow sphere") around the mean, a structure visible even in 2D PCA projections. **(e)** As predicted by theory, this concentration tightens relative to the dimension ($O(d^{-1/2})$).

2023a). This structure makes simple distance metrics effective, resolving the representation degeneration that plagued previous methods (Ma & Zhu, 2022; Ethayarajh, 2019). Building on this, research reveals that pre-trained models are often superior OOD detectors because they form clean domain-level clusters that task-specific fine-tuning fragments, inadvertently hiding OOD samples in the resulting gaps (Uppaal et al., 2023). This trade-off, which we term the *fine-tuning paradox*, is now being formalized by theoretical work connecting OOD robustness to PAC learning guarantees and the information-theoretic concept of a 'typical set' (Cover & Thomas, 1999).

OOD detection in LLMs follows three main paradigms, each with significant trade-offs. **Likelihood-based methods** use ratios between base and fine-tuned models as an OOD signal (Zhang et al., 2024; Ren et al., 2022), but are computationally prohibitive and assume the base model covers all anomalies. **Representation-based methods** exploit embedding geometry via distance metrics (Podolskiy et al., 2021) or lightweight PEFT activations (Hayou et al., 2024), but face a paradox where the fine-tuning needed for tasks degrades the geometric structure required for detection. Finally, **synthetic data generation** implements Outlier Exposure (Hendrycks et al., 2018) by creating challenging outliers (Abbas et al., 2025; Chen et al., 2021; Fort et al., 2021); however, this approach remains reactive, requiring an "OOD oracle" to anticipate threats and thus failing against unknown-unknowns.

The connection between OOD detection and LLM safety is direct: adversarial prompts, including jailbreaks , prompt injection (Liu et al., 2023b), and role-playing exploits (Wei et al., 2023), are by definition *out-of-distribution*, as they must deviate from natural language. This contrasts with dominant reactive defenses, such as specialized classifiers (Inan et al., 2023) or alignment techniques like RLHF (Ouyang et al., 2022) and Constitutional AI (Bai et al., 2022), which cannot generalize to novel threats and consistently lag the evolving adversarial landscape. While recent proactive work has begun applying OOD principles to address safety problems like anomaly detection, perplexity filtering (Jain et al., 2023), hallucination detection, and uncertainty quantification (Kuhn et al., 2023; Kadavath et al., 2022), a unified framework is still lacking.

Our work synthesizes these insights into a unified framework that resolves their fundamental limitations. T3 operationalizes the principle that safety is fundamentally about typicality (Nalisnick et al., 2019; Cover & Thomas, 1999) by learning the distribution of safe usage directly from curated examples. This approach avoids the high cost of dual-model likelihood methods (Zhang et al., 2024), preserves the clean geometric structure that fine-tuning corrupts (Uppaal et al., 2023), and requires no "OOD oracle" to anticipate threats as synthetic data methods do (Abbas et al., 2025). By adapting a principled OOD framework from vision (Ganguly et al., 2025c) to leverage the unique isotropic geometry of LLM embeddings (Liu et al., 2023a), we provide a proactive defense that is both theoretically grounded and efficient.

## 3 METHODOLOGY

**Problem Formulation:** We consider the problem of detecting potentially harmful prompts and LLM outputs before processing further. Let $\mathcal{D}_{\text{safe}}$ denote the distribution of benign prompts representing acceptable model usage. Given a reference corpus $X = \{x_i\}_{i=1}^m \sim \mathcal{D}_{\text{safe}}^m$ and test prompts $Y = \{y_j\}_{j=1}^n$, our goal is to determine whether each $y_j \sim \mathcal{D}_{\text{safe}}$ (in-distribution) or $y_j \sim \mathcal{D}_{\text{harmful}}$ (out-

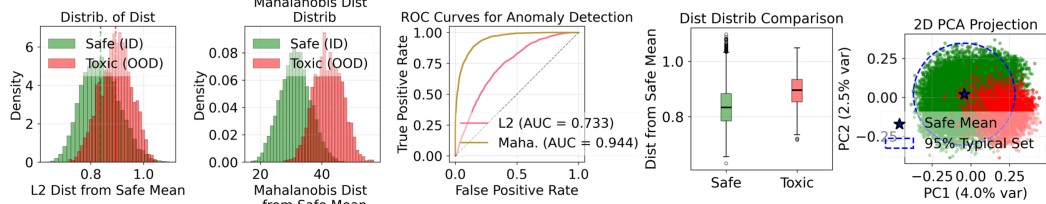

Figure 2: **Distinguishing safe vs. toxic text using geometric typicality.** This figure compares simple Euclidean and Mahalanobis distances for separating 10,000 safe and 2,000 toxic embeddings. **(a, b)** Mahalanobis distance, which accounts for the safe data's covariance, provides far better separation between safe (green) and toxic (red) distributions. **(c, d)** This superiority is quantified by a significantly higher ROC AUC (0.944 vs. 0.733) and confirmed by box plots. **(e)** A 2D PCA projection visually confirms that toxic samples fall predominantly outside the 95% typical set boundary of safe data.

of-distribution), where $\mathcal{D}_{\text{harmful}}$ represents an unknown distribution of adversarial prompts, toxic content, or off-topic queries. To this end, *we adapt the multi-model framework of Ganguly et al. (2025c) from visual to textual domain.* For each text sample $x$, we employ 3 sentence transformers: $\mathcal{E}_1$ (Qwen3-Embedding-0.6B), $\mathcal{E}_2$ (BGE-M3), and $\mathcal{E}_3$ (E5-Large-v2). Each encoder $\mathcal{E}_k$ produces a normalized embedding: $\phi_k(x) = \frac{\mathcal{E}_k(x)}{\|\mathcal{E}_k(x)\|_2} \in \mathbb{S}^{d_k-1}$ where $\mathbb{S}^{d_k-1}$ denotes the unit hypersphere in $\mathbb{R}^{d_k}$. This normalization ensures cosine similarity computations and mitigates encoder-specific scaling artifacts.

For each encoder and test point $y_j$, we compute four geometric features that capture the relationship between test and reference distributions. Let $\text{NB}_k(z; Z)$ denote the smallest ball centered at $z$ containing its $k$ nearest neighbors from set $Z$, and define the reference manifold estimate: $S_k(X) = \bigcup_{i=1}^m \text{NB}_k(\phi_k(x_i); X)$. The per-point PRDC metrics are: $\text{Precision}_k^{(j)} = \mathbb{1}(y_j \in S_k(X))$; $\text{Recall}_k^{(j)} = \frac{1}{m}\sum_{i=1}^m \mathbb{1}(\phi_k(x_i) \in \text{NB}_k(\phi_k(y_j); Y))$; $\text{Density}_k^{(j)} = \frac{1}{km}\sum_{i=1}^m \mathbb{1}(\phi_k(y_j) \in \text{NB}_k(\phi_k(x_i); X))$; $\text{Coverage}_k^{(j)} = \mathbb{1}(\exists i : \phi_k(x_i) \in \text{NB}_k(\phi_k(y_j); Y))$. These metrics have useful mathematical properties that enable principled anomaly detection. Under the null hypothesis $H_0 : \mathcal{D}_{\text{test}} = \mathcal{D}_{\text{safe}}$, we can compute the expected values of the metrics as follows.

**Theorem 3.1** (Expected Values under the null hypothesis). When test and reference samples are drawn from the same distribution:

$$\mathbb{E}[\text{Recall}_k^{(j)}] = k/n \qquad\qquad \mathbb{E}[\text{Density}_k^{(j)}] = 1/m$$
$$\mathbb{E}[\text{Coverage}_k^{(j)}] \leq 1 - (1 - k/n)^m \qquad \lim_{m \to \infty} \mathbb{E}[\text{Precision}_k^{(j)}] = 1$$

While the values of these metrics are analytically intractable without additional information about the distributions when $\mathcal{D}_{\text{safe}} \neq \mathcal{D}_{\text{test}}$, we compute the values in a number of interesting regimes and show that the metrics are *consistent tests* in these regimes, i.e. they can be used to distinguish the null hypothesis from the alternative hypothesis. We prove these results as well as the theorem above in Appendix A.

1. **Partial Support Mismatch**: When $\mathcal{D}_{\text{harmful}}(\text{supp}(\mathcal{D}_{\text{safe}})^c) = \alpha > 0$, harmful prompts explore semantic regions outside typical usage, yielding $\lim_{m \to \infty} \mathbb{E}[\text{Precision}] = 1 - \alpha < 1$.

2. **Density Shift**: Even when supports coincide, if density ratio $r(y) = p_{\text{safe}}(y)/p_{\text{harmful}}(y)$ is non-constant, coverage satisfies:

$$\lim_{m,n \to \infty} \mathbb{E}[\text{Coverage}] = 1 - \mathbb{E}_{y \sim \mathcal{D}_{\text{harmful}}}[e^{-\lambda k r(y)}] < 1 - e^{-\lambda k} \tag{1}$$

   where $\lambda = \lim_{m,n \to \infty} m/n$. This guarantees coverage is maximized only when distributions match.

3. **Local Perturbations**: For regions where $r(y) \leq 1 - \eta$ with $\mathcal{D}_{\text{harmful}}$-measure $\delta > 0$, the coverage gap is at least $\delta(e^{-\lambda k} - e^{-\lambda k(1-\eta)})$, ensuring detection of targeted adversarial patterns.

These results show that PRDC metrics are sufficiently powerful to capture the differences in two distributions, but do not directly give us the threshold values of the metrics for distinguishing two distributions. We use density estimation methods, trained only on the PRDC metrics computed from the In-Distribution data, to compute anomaly scores. We aggregate PRDC features across all encoders to form a single representation: $\mathbf{T}(y_j) = [\text{PRDC}_1^{(j)}, \ldots, \text{PRDC}_K^{(j)}] \in \mathbb{R}^{4K}$; where $\text{PRDC}_k^{(j)} = [\text{Recall}_k^{(j)}, \text{Density}_k^{(j)}, \text{Precision}_k^{(j)}, \text{Coverage}_k^{(j)}]$. This multi-view representation captures semantic anomalies that may be subtle in individual embedding spaces. We model the distribution of $\mathbf{T}$ on safe data using two complementary approaches: **Gaussian Mixture Model (GMM)** with components selected via Bayesian Information Criterion, and **One-Class SVM (OCSVM)** with RBF kernel with $\nu$ parameter tuned via validation accuracy. The anomaly score for test point $y_j$ is computed as the negative log-likelihood under the fitted model, with scores normalized to $[0, 1]$ via sigmoid transformation.

We contextualize the per-point PRDC metrics within the broader literature on non-parametric, $k$-nearest-neighbor–based two-sample testing. Classical pooled-graph tests ask the global question "do $F$ and $G$ match?"; by contrast, Forte tackles the harder, per-sample question of whether each $y_j$ is compatible with $X$, and it is intentionally asymmetric and scalable (reusing structure from the reference set). This asymmetry and the use of in-set rather than pooled neighbors mean these procedures are not equivalent in general, and naïvely adapting pooled tests for repeated prediction would be computationally prohibitive at modern scales. Even so, viewing PRDC and Forte or T3 through the two-sample-test lens clarifies their mathematical behavior and suggests useful sanity checks. The details of this comparison are given in Appendix A.

We evaluate performance using Area under ROC curve **(AUROC)**, measuring ranking quality across all thresholds, False positive rate at 95% true positive rate **(FPR@95)**, important for safety-sensitive applications, Area Under the Perturbation Recall Curve **(AUPRC)**, and Maximum F1 score with corresponding threshold, balancing precision and recall **(Optimal F1)**.

**Implementation Details.** For PRDC computation, we randomly split the reference set into two equal halves to avoid self-similarity bias when computing nearest neighbor statistics. The L2 distance on normalized embeddings is used throughout, exploiting the relationship $\|x - y\|_2^2 = 2(1 - \cos(x, y))$ for unit vectors. Embeddings are cached to disk in PyTorch format, enabling efficient reuse across experiments. The detector selection uses grid search over hyperparameters: GMM components $\in \{1, 2, 4, 8, 16, 32, 64\}$ constrained by sample size, and OCSVM $\nu \in \{0.01, 0.05, 0.1, 0.2, 0.5\}$.

## 4  RESULTS

We conduct a comprehensive empirical evaluation, aiming to answer five critical research questions:

**In-Distribution Data.** Following established OOD detection protocols (Hendrycks et al., 2019; Liang et al., 2022), we construct our in-distribution (ID) dataset from a curated mix of safe, helpful prompts spanning diverse domains. Our ID corpus combines high-quality instruction-following data from Alpaca (Taori et al., 2023), Dolly (Conover et al., 2023), and OpenAssistant (Köpf et al., 2023), equally distributed across the datasets, totaling approximately 40K examples. Critically, no harmful, toxic, or adversarial examples are included in the ID data for any OOD detector, ensuring a fair test of generalization.

**Out-of-Distribution Benchmarks.** We evaluate on 12 challenging OOD benchmarks representing the spectrum of LLM safety threats: general toxicity detection (RealToxicityPrompts (Gehman et al., 2020), CivilComments (Borkan et al., 2019)), targeted hate speech (HatEval (Basile et al., 2019), Davidson (Davidson et al., 2017), HASOC (Mandl et al., 2019), OffensEval (Zampieri et al., 2019)), multilingual harms (XSafety (Wang et al., 2023)), and domain-specific policy violations across code, cybersecurity, education, HR, and social media contexts (PolyGuard (Kang et al., 2025)). Additionally, we use four adversarial benchmarks: AdvBench (Zou et al., 2023), HarmBench (Mazeika et al., 2024), JailbreakBench (Chao et al., 2025), and MaliciousInstruct (Huang et al., 2023). Results on WildGuardMix (Han et al., 2024) are provided in Table 9.

**Baselines.** We compare against three categories of state-of-the-art methods: (1) *Specialized Safety Models*: LlamaGuard 3-1B and 4-12B (Inan et al., 2023), WildGuard (Han et al., 2024), DuoGuard (Deng et al., 2025), MD-Judge (Li et al., 2024), PolyGuard (Kang et al., 2025), and LLM-Guard

(Mhatre et al., 2025); (2) *Commercial Safety APIs*: OpenAI Omni Moderation and Perspective API; (3) *Representation-based OOD Methods*: We adapt ten established techniques to operate on semantic embeddings from Qwen3-Embedding-0.6B (Bai et al., 2023); ablations with larger embeddings (4B, 8B) are provided in Tables 10 and 11 ; text-native OOD baselines (Energy, kNN, Mahalanobis) are evaluated in Table 12, RMD (Ren et al., 2022), VIM (Wang et al., 2022), CIDER (Ming et al., 2022), GMM (Lee et al., 2018), OpenMax (Bendale & Boult, 2016), ReAct (Sun et al., 2021), AdaScale (Regmi, 2025), NNGuide (Park et al., 2023), and FDBD (Liu & Qin, 2023).

**Evaluation Metrics.** We report AUROC, AUPRC, F1-score, and FPR@95TPR (False Positive Rate at 95% True Positive Rate) (Hendrycks et al., 2019). For safety applications, FPR@95TPR is particularly critical as it measures the rate of false alarms while maintaining high detection sensitivity. Concretely, FPR@95TPR answers: "If we require catching 95% of harmful prompts, what fraction of truly safe prompts are mistakenly flagged?" For each benchmark, OOD (harmful) examples come from the respective dataset while ID (safe) examples come from our held-out safe corpus; we compute ROC curves over all scores and report FPR at the threshold achieving 95% TPR on OOD. Importantly, no OOD labels or test data are used to train T3's density estimators (GMM/OCSVM), and thresholds are not tuned per-benchmark—there is no data leakage from evaluation back into training.

**RQ1: How does T3 compare against specialized safety models and traditional OOD methods on diverse harm detection benchmarks?**

Table 1: Toxicity Detection Performance Across Six Benchmarks. T3 outperforms most baselines, including specialized safety models and traditional OOD methods. Performance is measured by AUROC (higher is better) and FPR@95 (lower is better). T3 achieves exceptionally low False Positive Rate (FPR@95), indicating high precision for practical deployment.

| Dataset | Civil Comments | | Davidson et al. | | Hasoc | | Hateval | | OffensEval | | Real Toxicity | |
| Metric | AUROC | FPR@95 | AUROC | FPR@95 | AUROC | FPR@95 | AUROC | FPR@95 | AUROC | FPR@95 | AUROC | FPR@95 |
| Method | | | | | | | | | | | | |
|---|---|---|---|---|---|---|---|---|---|---|---|---|
| ADASCALE | 0.3572 | 0.9971 | 0.1063 | 0.9999 | 0.4323 | 0.9925 | 0.3491 | 0.9890 | 0.2766 | 0.9987 | 0.5072 | 0.9672 |
| CIDER | 0.7393 | 0.9267 | 0.6791 | 0.9790 | 0.7880 | 0.8769 | 0.7827 | 0.8823 | 0.7469 | 0.9525 | 0.7535 | 0.8726 |
| FDBD | 0.4903 | 0.9921 | 0.7694 | 0.8960 | 0.4298 | 0.9941 | 0.5357 | 0.9730 | 0.6009 | 0.9674 | 0.3519 | 0.9944 |
| GMM | 0.6249 | 0.9758 | 0.6297 | 0.9757 | 0.6723 | 0.9609 | 0.7027 | 0.9689 | 0.7284 | 0.9637 | 0.6297 | 0.9565 |
| NNGUIDE | 0.4710 | 0.9960 | 0.2493 | 0.9996 | 0.5527 | 0.9949 | 0.4665 | 0.9849 | 0.4101 | 0.9995 | 0.6094 | 0.9600 |
| OPENMAX | 0.5347 | 0.9958 | 0.7966 | 0.9997 | 0.4644 | 0.9908 | 0.5874 | 0.9922 | 0.6042 | 0.9976 | 0.4396 | 0.9633 |
| REACT | 0.3432 | 0.9962 | 0.2016 | 0.9996 | 0.3925 | 0.9913 | 0.3784 | 0.9881 | 0.3059 | 0.9992 | 0.5536 | 0.9485 |
| RMD | 0.5560 | 0.9827 | 0.5989 | 0.9814 | 0.5982 | 0.9697 | 0.6123 | 0.9798 | 0.6529 | 0.9666 | 0.5635 | 0.9750 |
| VIM | 0.5626 | 0.9953 | 0.4742 | 0.9985 | 0.6046 | 0.9918 | 0.5776 | 0.9940 | 0.5967 | 0.9958 | 0.6601 | 0.9642 |
| | | | | | | | | | | | | |
| Perspective API | **0.9711** | **0.1413** | 0.9786 | 0.1065 | **0.9482** | 0.4062 | 0.9376 | 0.4070 | 0.9208 | 0.5171 | **0.9372** | 0.5106 |
| OpenAI Omni | 0.8916 | 0.8607 | 0.8926 | 0.9068 | 0.8591 | 0.9144 | 0.8861 | 0.8608 | 0.8069 | 0.9668 | 0.7557 | 0.9736 |
| | | | | | | | | | | | | |
| LLAMAGUARD3-1B | 0.5714 | 1.0000 | 0.6234 | 1.0000 | 0.5881 | 1.0000 | 0.7475 | 1.0000 | 0.6027 | 1.0000 | 0.5858 | 1.0000 |
| LLAMAGUARD4-12B | 0.5368 | 1.0000 | 0.5547 | 1.0000 | 0.5483 | 1.0000 | 0.6768 | 1.0000 | 0.5496 | 1.0000 | 0.5224 | 1.0000 |
| LLAMAGUARD3-1B-LOGITS | 0.7389 | 0.8214 | 0.8378 | 0.6565 | 0.7399 | 0.8261 | 0.8995 | 0.4861 | 0.8217 | 0.7004 | 0.7632 | 0.7820 |
| WILDGUARD | 0.7994 | 1.0000 | 0.8514 | 1.0000 | 0.7707 | 1.0000 | 0.8191 | 1.0000 | 0.7945 | 1.0000 | 0.6655 | 1.0000 |
| MDJUDGE | 0.7439 | 0.8552 | 0.7797 | 0.7926 | 0.7447 | 0.8397 | 0.7926 | 0.8201 | 0.8176 | 0.8746 | 0.6665 | 0.9186 |
| DUOGUARD | 0.8789 | 0.6742 | 0.8947 | 0.6170 | 0.8240 | 0.8230 | 0.7885 | 0.8119 | 0.8269 | 0.7516 | 0.7934 | 0.9110 |
| POLYGUARD | 0.7904 | 0.5446 | 0.8791 | 0.2216 | 0.7884 | 0.5206 | 0.8879 | 0.2593 | 0.7832 | 0.5315 | 0.7353 | 0.5380 |
| T3+GMM | 0.9249 | 0.2079 | **0.9869** | 0.0366 | 0.9198 | 0.2022 | 0.9809 | 0.0451 | 0.9886 | 0.0253 | 0.9282 | **0.1808** |
| T3+OCSVM | 0.9678 | 0.1722 | **0.9913** | 0.0350 | 0.9632 | 0.1860 | **0.9895** | 0.0408 | **0.9940** | 0.0201 | 0.9684 | 0.1670 |

Across six toxicity and hate speech benchmarks, T3 decisively outperforms all baselines, particularly in reducing false alarms. Our findings show that **traditional OOD methods fail catastrophically** when applied to semantic safety, with most exhibiting false positive rates (FPR@95) exceeding 90%, rendering them unusable. While **specialized safety models** like DuoGuard and PolyGuard achieve better detection (AUROC), they hit a **"precision ceiling,"** suffering from prohibitively high false positive rates (e.g., 75.2% for DuoGuard on OffensEval and 61.7% on Davidson) due to their reliance on reactive pattern-matching. In stark contrast, T3 **achieves order-of-magnitude improvements** in both detection and precision. T3-OCSVM delivers near-perfect AUROC ($\geq 0.96$ on 5 of 6 benchmarks) and, most critically, slashes false positives. On OffensEval, T3's 2.0% FPR@95 represents a **37× reduction** over the best baseline, with similar dramatic gains across all datasets. This stable, high-precision performance demonstrates the superiority of T3's proactive approach, which models the "typical set" of safe content rather than attempting to enumerate all possible harms. (see Table 6 for component ablations).

For LlamaGuard, we evaluate both the standard discrete classification and a fine-grained **logit-based scoring** variant (LLAMAGUARD3-1B-LOGITS). The logit-based approach extracts $p(\text{safe}) = \exp(\ell_{\text{safe}})/(\exp(\ell_{\text{safe}}) + \exp(\ell_{\text{unsafe}}))$ from the model's output logprobs, providing a continuous

confidence score rather than a binary decision. This more favorable scoring improves LlamaGuard's calibration and reduces its FPR@95TPR; however, T3 still achieves substantially better performance across all benchmarks.

**RQ2: Can T3, trained only on safe data, generalize to detect novel, unseen adversarial and jailbreaking attacks?**

Against six diverse adversarial and jailbreaking benchmarks, T3 provides a substantially more robust defense than existing methods despite being trained only on safe data. **Traditional OOD techniques again fail catastrophically**, proving useless for practical defense with false positive rates (FPR@95) typically exceeding 97%. **Specialized safety models exhibit attack-specific vulnerabilities** and inconsistent protection; even the strongest baseline, PolyGuard, still incorrectly flags over 64% of safe prompts on every benchmark. In contrast, T3's attack-agnostic approach of identifying statistical deviations delivers significant gains. It excels against **direct attacks**, reducing the FPR@95 on AdvBench to 15.8%, a **4.2× improvement** over PolyGuard, and performs well against **contextual manipulations.** While more **subtle attacks** remain challenging for all methods, T3's graduated response to threat sophistication, unlike the binary failures of other models, marks a significant step toward a more generalizable and practical adversarial defense.

**RQ3: How effectively does T3 mitigate the common problem of overrefusal on benign-but-challenging prompts? How is the cold-start performance with limited ID data?**

On the OR-Bench, designed to measure overrefusal on safe-but-challenging prompts, T3 provides the best balance between safety and utility. While traditional OOD methods fail by flagging most benign edge cases as harmful (FPR@95 >68%), specialized models show inconsistent performance. Llama-Guard achieves **an impressive low 14.8% FPR@95** on this specific task, a result that sharply contrasts its moderate performance on other benchmarks and suggests dataset-specific overfitting. Other models like DuoGuard and PolyGuard still over-refuse excessively (43.5% and 53.2% FPR@95). T3 delivers the most robust and well-rounded solution, with T3-GMM achieving an excellent **22.2% FPR@95** and T3-OCSVM the highest AUROC (0.934). We also evaluated an LLM-enhanced variant (denoted "Augment" in Table 3) that prepends a structured safety analysis from GPT-OSS-20B before embedding; however, this *decreases* OR-Bench performance, likely because the LLM's explicit safety labels shift borderline-safe prompts toward the harmful distribution in embedding space. Furthermore, T3 is highly sample-efficient and does not suffer from a cold start problem. As shown in Figure 3, performance converges rapidly with a small number of in-distribution examples. With just 500 safe samples, T3-OCSVM already achieves high AUROC and significantly reduced FPR@95 across most benchmarks.

Table 2: **T3 provides zero-shot defense against adversarial and jailbreaking attacks.** Trained only on safe data, T3 demonstrates significantly better generalization against six diverse attack benchmarks. It provides a robust, attack-agnostic defense, in contrast to specialized models which show inconsistent, attack-specific vulnerabilities.

| Dataset | AdvBench | | BeaverTails | | HarmBench | | JailbreakBench | | MaliciousInstruct | | XSTest | |
|---|---|---|---|---|---|---|---|---|---|---|---|---|
| Metric
Method | AUROC | FPR@95 | AUROC | FPR@95 | AUROC | FPR@95 | AUROC | FPR@95 | AUROC | FPR@95 | AUROC | FPR@95 |
| **ADASCALE** | 0.5894 | 0.9827 | 0.2833 | 0.9986 | 0.4341 | 0.9900 | 0.3500 | 0.9829 | 0.2994 | 1.0000 | 0.2803 | 0.9952 |
| **CIDER** | 0.2963 | 1.0000 | 0.2777 | 0.9974 | 0.4799 | 0.9650 | 0.5966 | 0.9556 | 0.2580 | 1.0000 | 0.3345 | 1.0000 |
| **FDBD** | 0.5689 | 0.9750 | 0.7226 | 0.8754 | 0.6342 | 0.9200 | 0.6195 | 0.9317 | 0.8510 | 0.7100 | 0.8021 | 0.7810 |
| **GMM** | 0.2737 | 0.9981 | 0.2515 | 0.9972 | 0.4163 | 0.9900 | 0.5377 | 0.9625 | 0.2252 | 1.0000 | 0.2298 | 1.0000 |
| **NNGUIDE** | 0.3989 | 0.9981 | 0.2239 | 0.9994 | 0.4023 | 0.9850 | 0.3795 | 0.9863 | 0.1622 | 1.0000 | 0.2144 | 1.0000 |
| **OPENMAX** | 0.3681 | 0.9942 | 0.6226 | 0.9984 | 0.4904 | 0.9900 | 0.5861 | 0.9898 | 0.5954 | 1.0000 | 0.6308 | 0.9952 |
| **REACT** | 0.4461 | 0.9962 | 0.2655 | 0.9980 | 0.3648 | 1.0000 | 0.2570 | 1.0000 | 0.1913 | 1.0000 | 0.2373 | 1.0000 |
| **RMD** | 0.3575 | 0.9981 | 0.3568 | 0.9727 | 0.4846 | 1.0000 | 0.4816 | 0.9898 | 0.3869 | 1.0000 | 0.3730 | 1.0000 |
| **VIM** | 0.3340 | 0.9981 | 0.2169 | 0.9997 | 0.3369 | 1.0000 | 0.4032 | 1.0000 | 0.1441 | 1.0000 | 0.1565 | 1.0000 |
| | | | | | | | | | | | | |
| **Perspective API** | 0.7895 | 0.9558 | 0.7922 | 0.8429 | 0.7247 | 0.9500 | 0.7233 | 0.9795 | 0.6828 | 1.0000 | 0.7932 | 0.7381 |
| **OpenAI Omni** | 0.8908 | 0.8269 | **0.8091** | 0.9488 | 0.8185 | 0.9650 | **0.8369** | 0.6724 | 0.8825 | 0.9200 | 0.8257 | 0.9667 |
| | | | | | | | | | | | | |
| **LLAMAGUARD3-1B** | 0.8894 | 1.0000 | 0.7135 | 1.0000 | 0.8857 | 1.0000 | 0.7248 | 1.0000 | 0.8507 | 1.0000 | 0.7843 | 1.0000 |
| **LLAMAGUARD3-1B-LOGITS** | 0.8110 | 0.7500 | 0.5598 | 0.9366 | **0.8887** | 0.3600 | 0.7293 | 0.6689 | 0.5791 | 0.9300 | 0.6542 | 0.9095 |
| **LLAMAGUARD4-12B** | 0.8822 | 1.0000 | 0.7137 | 1.0000 | **0.8868** | 1.0000 | 0.7165 | 1.0000 | 0.8718 | 1.0000 | 0.8120 | 1.0000 |
| **WILDGUARD** | 0.8658 | 1.0000 | 0.8218 | 1.0000 | 0.8642 | 1.0000 | 0.6978 | 1.0000 | 0.8617 | 0.9982 | 0.7929 | 1.0000 |
| **MDJUDGE** | 0.7814 | 0.9942 | 0.7779 | 0.8987 | 0.7980 | 0.9050 | 0.7302 | 0.8908 | 0.7957 | 0.9700 | 0.7906 | 0.9238 |
| **DUOGUARD** | 0.8241 | 0.9327 | **0.8525** | 0.8064 | 0.8007 | 0.9550 | 0.6820 | 0.9898 | 0.7745 | 1.0000 | 0.8418 | 0.7810 |
| **POLYGUARD** | 0.8670 | 0.6654 | 0.8071 | **0.7269** | 0.8595 | 0.6450 | 0.7904 | 0.7201 | 0.8501 | 0.7800 | 0.8007 | 0.8714 |
| **T3+GMM** | **0.9675** | **0.1577** | 0.7276 | **0.7847** | 0.7578 | 0.6700 | 0.7588 | 0.5358 | 0.8280 | **0.5900** | 0.6794 | 0.8143 |
| **T3+OCSVM** | **0.9578** | **0.1731** | 0.6081 | 0.8758 | 0.8102 | 0.5850 | **0.8622** | 0.4539 | 0.7586 | 0.6800 | 0.5800 | 0.8762 |

Table 3: **Performance on OR-Bench, a benchmark designed to measure overrefusal on safe-but-challenging prompts.** T3 achieves an excellent balance of safety and utility, with T3-GMM delivering a low FPR@95 while T3-OCSVM achieves the highest AUROC. This outperforms most specialized models like DuoGuard, though LlamaGuard shows an strong FPR@95 on this specific task.

Figure 3: **T3 is highly sample-efficient, avoiding the cold start problem.** T3's detection performance (AUROC) rapidly converges to $\approx 90\%$ with as few as 1000 in-distribution training samples, demonstrating its ability to learn the manifold of safe usage from a small, curated dataset.

| Method / Metric | AUROC | FPR@95 | AUPRC | F1 |
|---|---|---|---|---|
| RMD | 0.7474 | 0.7550 | 0.8674 | 0.8491 |
| VIM | 0.7162 | 0.7250 | 0.8386 | 0.8493 |
| CIDER | 0.7900 | 0.7117 | 0.9019 | 0.8536 |
| FDBD | 0.4169 | 0.9517 | 0.6662 | 0.8333 |
| NNGUIDE | 0.6220 | 0.9283 | 0.7862 | 0.8333 |
| REACT | 0.5438 | 0.9417 | 0.7420 | 0.8333 |
| GMM | 0.7530 | 0.6883 | 0.8679 | 0.8547 |
| ADASCALE | 0.5509 | 0.9783 | 0.7416 | 0.8333 |
| OPENMAX | 0.4710 | 0.9817 | 0.6871 | 0.8333 |
| | | | | |
| LLAMAGUARD3-1B | 0.8905 | 0.1483 | 0.9240 | 0.9346 |
| LLAMAGUARD4-12B | 0.8498 | 0.2748 | 0.9839 | 0.9796 |
| MDJUDGE | 0.8577 | 0.8900 | 0.9478 | 0.9082 |
| DUOGUARD | 0.9311 | 0.4350 | 0.9729 | 0.9063 |
| POLYGUARD | 0.8717 | 0.5317 | 0.9181 | 0.8833 |
| | | | | |
| T3+GMM | 0.9108 | 0.2217 | 0.9265 | 0.9405 |
| T3+GMM (Augment) | 0.8594 | 0.3267 | 0.9022 | 0.9153 |
| T3+OCSVM | 0.9342 | 0.2517 | 0.9662 | 0.9293 |
| T3+OCSVM (Augment) | 0.8581 | 0.4100 | 0.9114 | 0.9060 |

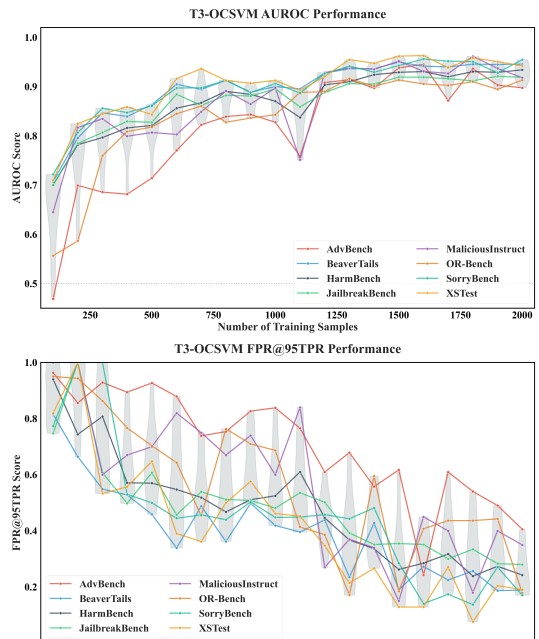

## RQ4: Does T3's performance generalize across different languages and specialized domains without retraining?

T3 demonstrates exceptional zero-shot generalization across specialized domains, using a single model trained only on general-purpose English prompts. Without any domain-specific training, T3 achieves **near-perfect, out-of-the-box performance**, with AUROC scores exceeding 99.5% and false positive rates (FPR@95) below 1% on both **Code** and **HR** policy violations, and similarly strong results in cybersecurity and education. In stark contrast, all baselines **fail to generalize**; traditional OOD methods are unusable (FPR@95 >93%), and specialized models like PolyGuard and LlamaGuard perform poorly even on their intended domains. This **40–100× improvement in FPR@95** validates T3's core principle: harmful content, whether it's malicious code or an HR violation, creates a consistent geometric signature of deviation from typical in-distribution language, enabling robust protection across diverse contexts without the need for retraining.

T3's zero-shot generalization extends powerfully across 14+ languages, from high-resource to lower-resource. Using only its English-trained model, T3 maintains **remarkably stable high-performance**, with T3-OCSVM showing less than 0.6% AUROC variance across all languages, including those with different scripts like Japanese and Arabic. This consistency starkly contrasts with specialized baselines like DuoGuard and PolyGuard, which exhibit **high linguistic variance** (up to 28%), making their performance unreliable across different regions. T3's success validates that harmful content creates a **language-agnostic geometric signature** in modern multilingual embedding spaces. This carries significant practical implications, as it eliminates the need for expensive multilingual data collection, retraining, and per-language calibration, enabling a single model to enforce a consistent safety standard globally.

**LLM-Enhanced Variant (Augment):** We also explored prepending a structured LLM-generated safety analysis (via GPT-OSS-20B) to each prompt before embedding. As shown in Table 5, this augmentation improves T3+GMM performance for non-English languages on RTP-LX (e.g., +1.7% AUROC for DE, +2.9% for ES) but degrades English and XSafety performance. Root cause analysis

revealed that the LLM often labels in the prompt's native language rather than English, reducing embedding consistency. This suggests LLM augmentation requires language-aware output normalization to be effective. Overall, the increase in overheads do not justify the increase in performance.

Table 4: Polyguard Domain-Specific Evaluation

| Dataset | Polyguard Code | | Polyguard Cyber | | Polyguard Education | | Polyguard HR | | Polyguard Social Media | |
| Metric | AUROC | FPR@95 | AUROC | FPR@95 | AUROC | FPR@95 | AUROC | FPR@95 | AUROC | FPR@95 |
| Method | | | | | | | | | | |
| ADASCALE | 0.7029 | 0.9665 | 0.6707 | 0.9484 | 0.4459 | 0.9976 | 0.4670 | 0.9978 | 0.2719 | 0.9997 |
| CIDER | 0.8919 | 0.3620 | 0.7425 | 0.8056 | 0.7406 | 0.9685 | 0.8299 | 0.9059 | 0.7645 | 0.9356 |
| FDBD | 0.2938 | 0.9978 | 0.3289 | 0.9974 | 0.5871 | 0.9896 | 0.6141 | 0.9951 | 0.6762 | 0.9480 |
| GMM | 0.8481 | 0.3709 | 0.7479 | 0.7549 | 0.6288 | 0.9804 | 0.7030 | 0.9826 | 0.6329 | 0.9858 |
| NNGUIDE | 0.8426 | 0.6939 | 0.6859 | 0.8855 | 0.5038 | 0.9940 | 0.5142 | 0.9974 | 0.4227 | 0.9995 |
| OPENMAX | 0.2899 | 0.9637 | 0.3410 | 0.9386 | 0.4712 | 0.9963 | 0.4304 | 0.9967 | 0.6126 | 0.9996 |
| REACT | 0.6079 | 0.9899 | 0.5077 | 0.9910 | 0.4478 | 0.9946 | 0.5151 | 0.9950 | 0.3205 | 0.9987 |
| RMD | 0.7358 | 0.7022 | 0.6701 | 0.8587 | 0.6505 | 0.9706 | 0.6864 | 0.9484 | 0.6185 | 0.9764 |
| VIM | 0.7926 | 0.6497 | 0.6453 | 0.9128 | 0.5022 | 0.9953 | 0.5153 | 0.9990 | 0.4606 | 0.9996 |
| LLAMAGUARD3-1B | 0.7139 | 1.0000 | 0.7789 | 1.0000 | 0.5740 | 1.0000 | 0.6368 | 1.0000 | 0.7482 | 1.0000 |
| LLAMAGUARD4-12B | 0.5235 | 1.0000 | 0.7733 | 1.0000 | 0.5389 | 1.0000 | 0.5520 | 1.0000 | 0.7151 | 1.0000 |
| WILDGUARD | 0.5706 | 1.0000 | 0.7463 | 1.0000 | 0.6637 | 1.0000 | 0.6833 | 1.0000 | 0.8252 | 1.0000 |
| LLAMAGUARD3-1B-LOGITS | 0.8031 | 0.7235 | 0.7519 | 0.7789 | 0.8661 | 0.6339 | 0.8257 | 0.7417 | 0.8249 | 0.6576 |
| MDJUDGE | 0.6491 | 0.8827 | 0.7616 | 0.8735 | 0.6909 | 0.8858 | 0.7146 | 0.8807 | 0.7445 | 0.8782 |
| DUOGUARD | 0.5356 | 0.8844 | 0.7574 | 0.8307 | 0.6626 | 0.9931 | 0.6363 | 0.9909 | 0.7224 | 0.9446 |
| POLYGUARD | 0.5530 | 0.7475 | 0.7354 | 0.8116 | 0.4464 | 0.9558 | 0.4198 | 0.9484 | 0.7224 | 0.6808 |
| T3+GMM | **0.9959** | **0.0089** | **0.9886** | **0.0270** | **0.9913** | **0.0255** | **0.9965** | **0.0062** | **0.9673** | **0.1208** |
| T3+OCSVM | **0.9953** | **0.0095** | **0.9818** | **0.0615** | **0.9943** | **0.0192** | **0.9982** | **0.0039** | **0.9620** | **0.1485** |

Table 5: **Consistent and stable performance across 14+ languages.** T3 maintains exceptionally high AUROC with minimal variance (<2%) across high- and low-resource languages, demonstrating its language-agnostic safety capabilities. Results are shown for the RTP LX (top) and XSafety (bottom) benchmarks.

| Dataset=RTP_LX | De | En | Es | Fr | Hi | It | Ja | Ko | Nl | Pl | Pt | Ru | Tr | Zh |
| LLAMAGUARD3-1B | 0.7746 | 0.7865 | 0.7997 | 0.7647 | 0.7877 | 0.7696 | 0.7715 | 0.7802 | 0.7990 | 0.7452 | 0.7677 | 0.7627 | 0.7571 | 0.8302 |
| MDJUDGE | 0.8617 | 0.8832 | 0.8718 | 0.8458 | 0.7868 | 0.8332 | 0.8673 | 0.8343 | 0.8418 | 0.7874 | 0.8544 | 0.8154 | 0.7865 | 0.9001 |
| DUOGUARD | **0.9876** | **0.9886** | **0.9884** | **0.9925** | 0.9521 | 0.9714 | 0.9682 | 0.8850 | **0.9785** | 0.9004 | **0.9817** | 0.8254 | 0.9351 | **0.9924** |
| POLYGUARD | 0.9551 | **0.9818** | 0.9660 | 0.9533 | 0.8396 | 0.9328 | **0.9779** | 0.9533 | 0.9379 | 0.8608 | 0.9548 | **0.9769** | 0.8515 | **0.9898** |
| T3+GMM | 0.9588 | 0.9554 | 0.9522 | 0.9546 | 0.9535 | 0.9540 | 0.9605 | 0.9600 | 0.9555 | 0.9550 | 0.9560 | 0.9572 | 0.9604 | 0.9526 |
| T3+GMM (Augment) | 0.9759 | 0.9071 | **0.9816** | 0.9756 | **0.9789** | **0.9738** | 0.9771 | **0.9871** | 0.9695 | **0.9728** | 0.9699 | 0.9741 | **0.9769** | 0.984 |
| T3+OCSVM | **0.9788** | 0.9804 | 0.9797 | **0.9807** | **0.9787** | **0.9805** | **0.9819** | **0.9811** | **0.9790** | **0.9806** | **0.9821** | **0.9768** | **0.9816** | 0.9812 |
| T3+OCSVM (Augment) | 0.9507 | 0.7909 | 0.9496 | 0.9492 | 0.9421 | 0.9491 | 0.9514 | 0.9606 | 0.96 | 0.9573 | 0.9495 | 0.952 | 0.9547 | 0.9619 |

| Dataset=XSafety | De | En | Es | Fr | Hi | | Ja | | | | | Ru | | Zh | Ar |
| LLAMAGUARD3-1B | 0.6215 | 0.6452 | 0.6383 | 0.6477 | 0.6421 | | 0.6183 | | | | | 0.6433 | | 0.6302 | 0.6633 |
| MDJUDGE | 0.7905 | 0.7765 | 0.8056 | 0.8003 | 0.7584 | | 0.7993 | | | | | 0.7874 | | 0.7886 | 0.7760 |
| DUOGUARD | 0.9228 | 0.7820 | 0.9085 | 0.9295 | **0.9693** | | 0.8357 | | | | | 0.7877 | | 0.8837 | 0.8286 |
| POLYGUARD | 0.7653 | 0.7051 | 0.7499 | 0.7682 | 0.8279 | | 0.7852 | | | | | 0.7811 | | 0.7354 | 0.8239 |
| T3+GMM | **0.9542** | **0.9476** | **0.9482** | **0.9602** | 0.9509 | | **0.9469** | | | | | **0.9542** | | **0.9522** | **0.9526** |
| T3+GMM (Augment) | 0.8942 | 0.9004 | 0.9740 | 0.8981 | 0.9102 | | 0.9003 | | | | | 0.9012 | | 0.9014 | 0.9026 |
| T3+OCSVM | **0.9815** | **0.9801** | **0.9762** | **0.9781** | **0.9797** | | **0.9804** | | | | | **0.9802** | | **0.9772** | **0.9800** |
| T3+OCSVM (Augment) | 0.7869 | 0.7802 | 0.9300 | 0.7786 | 0.786 | | 0.7907 | | | | | 0.7871 | | 0.7852 | 0.7887 |

## RQ5: Can T3 be integrated into a high-performance inference engine for practical, real-time deployment with minimal latency?

To demonstrate T3's practical deployment capabilities, we integrated it directly into the vLLM inference framework (Kwon et al., 2023) for real-time safety monitoring during generation. Unlike post-processing approaches that evaluate complete outputs, our system performs continuous safety assessment as tokens are generated, enabling immediate termination of harmful content. The integration leverages vLLM's multiprocess architecture by intercepting and accessing outputs in the mostly idling Main Process while inference proceeds in Worker Processes, achieving efficient computation overlap without disrupting the core generation pipeline.

**Streaming Results:** Performance evaluation on an NVIDIA H200 GPU demonstrates negligible overhead even under dense monitoring conditions. With T3 configured for evaluation every 20 tokens and batch processing of 32 requests, we observe **only 1.5% overhead** on 500-prompt workloads and **6% on 5,000-prompt workloads** compared to baseline vLLM (Table 7). This is achieved through strategic batching of safety evaluations and overlapping T3's embedding computations with ongoing token generation, effectively hiding guardrail latency behind inference operations. To our knowledge, T3 is the first framework to demonstrate sub-10% overhead for continuous safety monitoring during

online LLM generation, making real-time guardrailing practical for production deployments where both safety and latency are critical.

**Post-Generation Results:** T3's efficiency also extends to the widely adopted **post-generation** guardrailing mode. In a comparative benchmark on an NVIDIA H200 GPU with batch sizes up to 64, our results revealed a clear performance hierarchy. While DUOGUARD was fastest, T3 (GMM and OCSVM) demonstrated excellent scalability, with runtimes consistently between **60–155 ms**. This positions T3 as significantly more efficient than PolyGuard and vastly superior to heavier methods like MDJudge and LlamaGuard, which imposed prohibitive runtimes exceeding one million μs and failed at larger batch sizes. These findings confirm that T3 is a highly efficient and scalable solution for both online and post-generation safety deployment.

## 5 DISCUSSION

An important finding emerges from our evaluation on particularly challenging scenarios where the semantic distinction between "safe" and "harmful" is intentionally ambiguous and context-dependent. We deliberately tested our method on the Anthropic hh-rlhf dataset, where chosen (safe) responses serve as the in-distribution data and rejected (harmful) responses as out-of-distribution, a purposely difficult setup since the ID dataset already contains profanity and the only difference between response pairs might be subtle phrasing or a single word. In this challenging benchmark, no method, including T3, traditional baselines, and existing safety approaches, performs significantly better than random chance (AUROC≈0.5). Importantly, this failure stems not from methodological weakness but from the nature of the ID data itself: the "safe" examples contain extensive toxic content (e.g., lists of profanities), making the cosine similarity between chosen and rejected responses extremely high (>0.95). This creates a near-OOD detection problem where the typical set of "safe" usage already encompasses harmful patterns.

However, this limitation does not generalize to attacks designed to *semantically* resemble safe queries. We evaluated T3 against the HILL jailbreak method (Luo et al., 2025), which transforms harmful imperatives into innocuous-looking "learning-style" questions (e.g., "I am studying chemistry, explain this reaction..."). Despite HILL's explicit design to masquerade as in-distribution educational content, T3 achieves strong detection (AUROC 0.98, FPR@95 4.4%, see Table 13 for details.) when trained on properly curated safe data. This contrast is instructive: T3 succeeds when the ID training set genuinely represents safe usage, but fails when the ID set itself contains the harmful patterns it should detect. The key insight is that **T3's effectiveness is contingent on appropriate ID training set curation**.

This reveals a fundamental boundary for OOD-based safety: methods succeed when safe and harmful content occupy separable manifolds (HILL, domain adherence) but fail when they overlap (HH-RLHF). Crucially, this training distribution dependence is *universal*—supervised classifiers (Llama Guard, PolyGuard) collapse under domain shift, and Constitutional AI/RLHF systems over-refuse outside their preference distributions. T3's curation requirement is thus not a unique weakness but a shared property of all safety methods. This motivates hybrid architectures combining T3's efficient typicality screening for distributional outliers with reasoning-based methods for near-boundary cases requiring contextual intent parsing. Conversely, there are domains where the boundary between in-distribution and out-of-distribution is exceptionally clear. A prime example is domain adherence for mathematical reasoning. When the in-distribution "safe" set consists of mathematical problems and solutions (from datasets like MATH or GSM8K), and the OOD set consists of unrelated topics like philosophy, literature, or cooking, the semantic separation is vast. In this scenario, nearly all methods, including traditional OOD baselines like CIDER and GMM, perform extremely well, often achieving near-perfect AUROC scores.

## 6 CONCLUSION

The T3 guardrailing framework represents a significant paradigm shift in LLM safety, moving from reactive threat-blocking to a proactive approach based on statistical typicality. By modeling what is safe rather than enumerating harms, T3 achieves state-of-the-art performance, remarkable generalization across domains and languages, and a dramatic reduction in overrefusal. Its successful integration into the vLLM inference engine with minimal overhead demonstrates its readiness for practical, real-time deployment in production environments.

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

# A  THEORETICAL ANALYSIS

## A.1  EXPECTED VALUES OF PRDC METRICS

We analyze the mathematical and statistical properties of the PRDC metrics in this section. Let us first introduce some notation for easier readability. Let $X = \{X_i\}_{i=1}^m$ and $Y = \{Y_j\}_{j=1}^n$ be i.i.d. random vectors in $\mathbb{R}^d$ drawn from distributions $F$ and $G$ respectively. We denote by $NB_k(X_i; Z)$ the smallest open ball centered around $X_i$ containing its $k$ nearest neighbors from the set $Z$. For brevity, we write $NB_k(X_i)$ and $NB_k(Y_j)$ for $NB_k(X_i; X)$ and $NB_k(Y_j; Y)$ respectively. Let us recall the definitions of the per-point PRDC metrics introduced earlier, treating $X$ and the reference points and $Y$ as test points,

$$P_k^{(j)}(X,Y) = \mathbb{1}\left(Y_j \in \bigcup_{i=1}^m NB_k(X_i)\right)$$

$$R_k^{(j)}(X,Y) = \frac{1}{m}\sum_{i=1}^m \mathbb{1}\left(X_i \in NB_k(Y_j)\right)$$

$$D_k^{(j)}(X,Y) = \frac{1}{mk}\sum_{i=1}^m \mathbb{1}\left(Y_j \in NB_k(X_i)\right)$$

$$C_k^{(j)}(X,Y) = \mathbb{1}\left(\exists i, X_i \in NB_k(Y_j)\right)$$

In the following theorem, we compute the expected values of these metrics in the general case, without making any additional assumptions. Note that the expectation of the precision, $\mathbb{E}[P_k^{(j)}]$ is analytically intractable in general cannot be simplified any further without stronger assumptions.

**Theorem A.1.** Let $X = \{X_i\}_{i=1}^m$ and $Y = \{Y_j\}_{j=1}^n$ be sets of i.i.d. random vectors in $\mathbb{R}^d$ drawn from distributions $F$ and $G$ respectively. The expectations of the per-point metrics $R_k^{(j)}$, $D_k^{(j)}$, and $C_k^{(j)}$ are given by

1. $\mathbb{E}\left[R_k^{(j)}(X,Y)\right] = \mathbb{P}(X_1 \in NB_k(Y_1))$

2. $\mathbb{E}\left[D_k^{(j)}(X,Y)\right] = \frac{1}{k}\mathbb{P}(Y_1 \in NB_k(X_1))$

3. $\mathbb{E}\left[C_k^{(j)}(X,Y)\right] = 1 - \mathbb{E}\left[(1 - \mathbb{P}(X_1 \in NB_k(Y_1)))^m\right]$

the outer expectation in the third result is over the random sample $Y = \{Y_j\}_{j=1}^n$.

*Proof.* We prove each statement individually.

1. By definition, the expectation is:

$$\mathbb{E}\left[R_k^{(j)}(X,Y)\right] = \mathbb{E}\left[\frac{1}{m}\sum_{i=1}^m \mathbb{1}(X_i \in NB_k(Y_j))\right]$$

$$= \frac{1}{m}\sum_{i=1}^m \mathbb{E}\left[\mathbb{1}(X_i \in NB_k(Y_j))\right] \quad \text{(by linearity of expectation)}$$

$$= \frac{1}{m}\sum_{i=1}^m \mathbb{P}(X_i \in NB_k(Y_j)) \quad \text{(since } \mathbb{E}[\mathbb{1}(A)] = \mathbb{P}(A))$$

The random variables $\{X_i\}_{i=1}^m$ are i.i.d. from $F$, and $\{Y_j\}_{j=1}^n$ are i.i.d. from $Q$. Therefore, the probability $\mathbb{P}(X_i \in NB_k(Y_j))$ is identical for all choices of indices $i \in \{1, \ldots, m\}$ and

$j \in \{1, \ldots, n\}$. We can thus write this common probability as $\mathbb{P}(X_1 \in NB_k(Y_1))$.

$$\mathbb{E}\left[R_k^{(j)}(X, Y)\right] = \frac{1}{m} \sum_{i=1}^{m} \mathbb{P}(X_1 \in NB_k(Y_1))$$

$$= \frac{1}{m} \cdot m \cdot \mathbb{P}(X_1 \in NB_k(Y_1))$$

$$= \mathbb{P}(X_1 \in NB_k(Y_1))$$

2. The proof follows the same structure.

$$\mathbb{E}\left[D_k^{(j)}(X, Y)\right] = \mathbb{E}\left[\frac{1}{mk} \sum_{i=1}^{m} \mathbb{1}(Y_j \in NB_k(X_i))\right]$$

$$= \frac{1}{mk} \sum_{i=1}^{m} \mathbb{E}\left[\mathbb{1}(Y_j \in NB_k(X_i))\right] \quad \text{(by linearity of expectation)}$$

$$= \frac{1}{mk} \sum_{i=1}^{m} \mathbb{P}(Y_j \in NB_k(X_i))$$

Again, by the i.i.d. property of the samples $X$ and $Y$, the probability $\mathbb{P}(Y_j \in NB_k(X_i))$ is identical for all $i, j$. We write this common probability as $\mathbb{P}(Y_1 \in NB_k(X_1))$.

$$\mathbb{E}\left[D_k^{(j)}(X, Y)\right] = \frac{1}{mk} \sum_{i=1}^{m} \mathbb{P}(Y_1 \in NB_k(X_1))$$

$$= \frac{1}{mk} \cdot m \cdot \mathbb{P}(Y_1 \in NB_k(X_1))$$

$$= \frac{1}{k} \mathbb{P}(Y_1 \in NB_k(X_1))$$

3. The expectation of the indicator function is the probability of the underlying event.

$$\mathbb{E}\left[C_k^{(j)}(X, Y)\right] = E\left[\mathbb{1}(\exists i, X_i \in NB_k(Y_j))\right]$$

$$= \mathbb{P}\left(\bigcup_{i=1}^{m} \{X_i \in NB_k(Y_j)\}\right)$$

We use the law of total expectation by conditioning on the random sample $Y = \{Y_j\}_{j=1}^{n}$.

$$\mathbb{E}\left[C_k^{(j)}(X, Y)\right] = \mathbb{E}_Y\left[\mathbb{P}\left(\bigcup_{i=1}^{m} \{X_i \in NB_k(Y_j)\} \,\middle|\, Y\right)\right]$$

Conditioned on $Y$, the ball $NB_k(Y_j)$ is a fixed set. The events $\{X_i \in NB_k(Y_j)\}$ for $i = 1, \ldots, m$ are independent because the $X_i$ are i.i.d. and independent of $Y$. It is easier to compute the probability of the complement event,

$$\mathbb{P}\left(\bigcup_{i=1}^{m} \{X_i \in NB_k(Y_j)\} \,\middle|\, Y\right) = 1 - \mathbb{P}\left(\bigcap_{i=1}^{m} \{X_i \notin NB_k(Y_j)\} \,\middle|\, Y\right)$$

$$= 1 - \prod_{i=1}^{m} \mathbb{P}\left(X_i \notin NB_k(Y_j) \,|\, Y\right) \quad \text{(by conditional independence)}$$

The conditional probability $\mathbb{P}(X_i \in NB_k(Y_j)|Y)$ is the measure of the set $NB_k(Y_j)$ under the distribution $F$, which is the same for all $i$.

$$\mathbb{P}\left(\bigcup_{i=1}^{m} \{X_i \in NB_k(Y_j)\} \,\middle|\, Y\right) = 1 - \prod_{i=1}^{m} (1 - \mathbb{P}(X_1 \in NB_k(Y_j)))$$

$$= 1 - (1 - \mathbb{P}(X_1 \in NB_k(Y_j)))^m$$

Taking the expectation over $Y$ gives the final result.

$$\mathbb{E}\left[C_k^{(j)}(X,Y)\right] = \mathbb{E}_Y\left[1 - (1 - \mathbb{P}(X_1 \in NB_k(Y_j)))^m\right] = 1 - \mathbb{E}_Y\left[(1 - \mathbb{P}(X_1 \in NB_k(Y_j)))^m\right]$$

Since the $Y_j$ are i.i.d., the distribution of the random set $NB_k(Y_j)$ is the same for all $j$. We can therefore replace the index $j$ with 1 without loss of generality.

$$E\left[C_k^{(j)}(X,Y)\right] = 1 - E\left[(1 - \mathbb{P}(X_1 \in NB_k(Y_1)))^m\right]$$

$\square$

We note the important special case when $F = G$, i.e. the in-distribution setting when $X$ and $Y$ are drawn from the same distribution.

**Theorem A.2.** Let $X = \{X_i\}_{i=1}^m$ and $Y = \{Y_j\}_{j=1}^n$ be sets of i.i.d. random vectors in $\mathbb{R}^d$ drawn from the same distribution $F$. The expectations of the per-point metrics simplify to

1. $\mathbb{E}\left[R_k^{(j)}(X,Y)\right] = \dfrac{k}{n}$

2. $\mathbb{E}\left[D_k^{(j)}(X,Y)\right] = \dfrac{1}{m}$

3. $\mathbb{E}\left[C_k^{(j)}(X,Y)\right] \leq 1 - \left(1 - \dfrac{k}{n}\right)^m$

*Proof.* The assumption that $F = G$ implies that all $m + n$ vectors are i.i.d. samples from the same continuous distribution $F$. This allows us to use a symmetry argument.

1. From the general case, we know $\mathbb{E}\left[R_k^{(j)}(X,Y)\right] = \mathbb{P}(X_1 \in NB_k(Y_1))$. The event $\{X_1 \in NB_k(Y_1)\}$ means that the distance $\|X_1 - Y_1\|$ is less than the distance from $Y_1$ to its $k$-th nearest neighbor in the set $Y \setminus \{Y_1\} = \{Y_2, \ldots, Y_n\}$.

   Consider the set of $n$ points $\{X_1, Y_2, \ldots, Y_n\}$. Since $F = G$, these are $n$ i.i.d. samples from $F$. Let us consider their distances to the point $Y_1$. Since the distribution $F$ is continuous, the distances will be unique with probability 1. The set of distances $\{\|X_1 - Y_1\|, \|Y_2 - Y_1\|, \ldots, \|Y_n - Y_1\|\}$ consists of $n$ i.i.d. random variables.

   The event $\{X_1 \in NB_k(Y_1)\}$ is equivalent to the statement that $\|X_1 - Y_1\|$ is among the $k$ smallest values in this set of $n$ distances. By symmetry, any specific distance in the set is equally likely to have any rank from 1 to $n$. The probability that $\|X_1 - Y_1\|$ is one of the $k$ smallest is therefore

$$\mathbb{E}\left[R_k^{(j)}(X,Y)\right] = \mathbb{P}(X_1 \in NB_k(Y_1)) = \frac{k}{n}.$$

2. From the general case, $\mathbb{E}\left[D_k^{(j)}(X,Y)\right] = \frac{1}{k}\mathbb{P}(Y_1 \in NB_k(X_1))$. The logic is identical to the proof above, but with the roles of $X$ and $Y$ swapped. Consider the set of $m$ i.i.d. points $\{Y_1, X_2, \ldots, X_m\}$ and their distances to the point $X_1$. The event $\{Y_1 \in NB_k(X_1)\}$ is equivalent to the distance $\|Y_1 - X_1\|$ being among the $k$ smallest of the $m$ i.i.d. distances $\{\|Y_1 - X_1\|, \|X_2 - X_1\|, \ldots, \|X_m - X_1\|\}$.

   By symmetry, the probability of this event is

$$\mathbb{P}(Y_1 \in NB_k(X_1)) = \frac{k}{m}$$

   Substituting this into the expression for the expectation gives

$$\mathbb{E}\left[D_k^{(j)}(X,Y)\right] = \frac{1}{k} \cdot \frac{k}{m} = \frac{1}{m}.$$

3. The proof for this upper bound relies on Jensen's inequality. We begin by noting that a naive substitution of the average value of the probability mass of the k-NN ball would be incorrect. Specifically, the expectation of $C_k^{(j)}(X, Y)$ involves the non-linear function $f(z) = (1 - z)^m$. For such functions, the expectation of the function is generally not equal to the function of the expectation, i.e., $E[f(Z)] \neq f(E[Z])$.

The exact expression for the expectation is

$$E\left[C_k^{(j)}(X, Y)\right] = 1 - E\left[(1 - \mathbb{P}(X_1 \in NB_k(Y_1)))^m\right]$$

As stated above, $\mathbb{E}[\mathbb{P}(X_1 \in NB_k(Y_1))] = k/n$. Jensen's inequality states that for a convex function $f$ and a random variable $Z$, we have $E[f(Z)] \geq f(E[Z])$. The function in our case is $f(z) = (1 - z)^m$, which is a convex function. Applying Jensen's inequality,

$$E\left[(1 - \mathbb{P}(X_1 \in NB_k(Y_1)))^m\right] \geq \left(1 - \frac{k}{n}\right)^m$$

Finally, we substitute this inequality back into the expression for the expectation of $C_k^{(j)}(X, Y)$ to get

$$E\left[C_k^{(j)}(X, Y)\right] \leq 1 - \left(1 - \frac{k}{n}\right)^m.$$

$\square$

While the expression for $P_k^{(j)}(X, Y)$ is intractable in general even for the in-distribution case, its limiting value can still provide us some intuition about the metric. We now consider the asymptotic behavior of $\mathbb{E}[P_k^{(j)}(X, Y)]$ when both $X = \{X_i\}_{i=1}^m$ and $Y_j$ are drawn i.i.d. from the same distribution $F$ on $\mathbb{R}^d$, and the reference sample size $m$ tends to infinity. Let

$$S_m(X) = \bigcup_{i=1}^m NB_k(X_i)$$

denote the random set obtained from the sample $X$. Then

$$\mathbb{E}[P_k^{(j)}(X, Y)] = \mathbb{P}(Y_j \in S_m(X)).$$

Assume that $F$ has compact support $\mathrm{supp}(F)$, is absolutely continuous with density $f$ that is bounded and bounded away from zero on $\mathrm{supp}(F)$, and that the boundary $\partial\mathrm{supp}(F)$ has measure zero. Under these standard regularity conditions, nonparametric set estimation results imply that for fixed $k \geq 1$, $S_m(X) \to \mathrm{supp}(F)$ in probability (e.g. in Hausdorff distance), as $m \to \infty$. Intuitively, as the sample becomes dense, the $k$-NN radii shrink uniformly, so the union of $k$-NN balls fills out the entire support.

By continuity of probability measures and the fact that $Y_j \sim F$,

$$\lim_{m\to\infty} \mathbb{P}(Y_j \in S_m(X)) = \mathbb{P}(Y_j \in \mathrm{supp}(F)).$$

Since $Y_j$ is drawn from $F$, it lies in $\mathrm{supp}(F)$ with probability one. Therefore,

$$\lim_{m\to\infty} \mathbb{E}[P_k^{(j)}(X, Y)] = 1.$$

In the in-distribution case, as the reference sample grows, the estimated manifold $S_m(X)$ converges to the true support of $F$. Consequently, any new sample $Y_j \sim F$ will eventually fall inside $S_m(X)$ with probability approaching 1.

## A.2 CONSISTENCY

A statistical test is said to be consistent if its probability of distinguishing the null hypothesis from any alternative hypothesis converges to 1 as the sample size increases. We consider a few different regimes in which the expectations of the per–point PRDC metrics differ between the in–distribution setting $F = G$ and the out-of-distribution setting $F \neq G$, i.e. regimes in which PRDC metrics are consistent tests. We consider the asymptotic regime $m, n \to \infty$ with fixed $k$ and assume that $\lim_{m,n\to\infty} m/n = \lambda \in (0, \infty)$.

**(1) Partial support mismatch** Assume $F$ has compact support $\text{supp}(F)$ and let
$$\alpha := G\big(\text{supp}(F)^c\big) > 0.$$
This says that there is some region where $F$ has zero probability whereas $G$ has non-zero probability. Under mild regularity conditions stated earlier (compact support, $F$ absolutely continuous with density bounded and bounded away from zero on $\text{supp}(F)$), the union $\bigcup_{i=1}^m NB_k(X_i)$ converges in probability to $\text{supp}(F)$ as $m \to \infty$. Hence, for $Y_j \sim G$,
$$\lim_{m \to \infty} \mathbb{E}\, P_k^{(j)}(X, Y) = G(\text{supp}(F)) = 1 - \alpha < 1$$
whereas in the in–distribution case $F = G$ we have $\lim_{m \to \infty} \mathbb{E}\, P_k^{(j)}(X, Y) = 1$. For coverage, using
$$\mathbb{E}\, C_k^{(j)}(X, Y) \;=\; \mathbb{E}_Y\Big[1 - \big(1 - F\big(NB_k(Y_j)\big)\big)^m\Big],$$
any $Y_j \notin \text{supp}(F)$ contributes $0$ for all $m$, whence
$$\limsup_{m,n \to \infty} \mathbb{E}\, C_k^{(j)}(X, Y) \;\leq\; (1 - \alpha) \limsup_{m,n \to \infty} \mathbb{E}\big[\, 1 - \big(1 - F(NB_k(Y_j))\big)^m \,\big|\, Y \in \text{supp}(F)\big],$$
where the expectation on the right hand side is the expectation in the case when $F = G$, resulting in a strictly lower value of ($\limsup$ of the expected value of) coverage whenever $\alpha > 0$.

**(2) Same support, different densities** Assume $\text{supp}(F) = \text{supp}(G) =: S$ and $F, G$ are absolutely continuous with respect to Lebesgue measure on $S$ with continuous densities $f, g$ that are bounded and bounded away from zero on $S$. Let
$$r(y) \;:=\; \frac{\mathrm{d}F}{\mathrm{d}G}(y) \;=\; \frac{f(y)}{g(y)} \qquad \text{so that} \quad \mathbb{E}_{Y_j \sim G}[r(Y_j)] = 1.$$
For fixed $k$ and $n \to \infty$, we have
$$F\big(NB_k(Y_j)\big) \;=\; \frac{k}{n}\, r(Y_j)\,\big(1 + o_p(1)\big).$$
Substituting into the coverage identity gives
$$\lim_{m,n \to \infty} \mathbb{E}\, C_k^{(j)}(X, Y) \;=\; 1 - \mathbb{E}_{Y_j \sim G}\Big[\exp\big(-\lambda k\, r(Y_j)\big)\Big]. \tag{2}$$
When $F = G$ we have $r \equiv 1$ and recover $1 - e^{-\lambda k}$. When $F \neq G$, $r$ is non–constant on a set of positive $G$–measure and, since $z \mapsto e^{-\lambda k z}$ is convex, Jensen's inequality is strict:
$$\mathbb{E}_G\big[e^{-\lambda k r(Y_j)}\big] \;>\; e^{-\lambda k\, \mathbb{E}[r(Y_j)]} \;=\; e^{-\lambda k},$$
so that
$$\lim_{m,n \to \infty} \mathbb{E}\, C_k^{(j)}(X, Y) \;<\; 1 - e^{-\lambda k}.$$
Thus coverage is maximized at $F = G$ and strictly smaller otherwise, providing consistency even when supports coincide.

**(3) Different densities in a small region** Within the same-support setting above, suppose there exist $\eta \in (0, 1)$ and a measurable set $A \subset S$ with $G(A) = \delta > 0$ such that
$$r(y) \leq 1 - \eta \qquad \text{for all } y \in A,$$
i.e. there is a set with positive $G$ probability where $F$ has strictly smaller density than $G$. Conditioning on $Y_j \in A$ vs. $Y_j \notin A$ in equation 2 gives
$$\lim_{m,n \to \infty} \mathbb{E}\, C_k^{(j)}(X, Y) \;\leq\; 1 - \Big(\delta\, e^{-\lambda k(1-\eta)} + (1 - \delta)\, e^{-\lambda k}\Big),$$
so the gap from the in–distribution baseline $1 - e^{-\lambda k}$ is at least
$$\delta\Big(e^{-\lambda k} - e^{-\lambda k(1-\eta)}\Big) \;>\; 0.$$
This captures a practically relevant situation where a nontrivial portion of the $G$–mass lies in regions with systematically lower $F$–density; coverage reflects this "margin" as a strict and quantifiable decrease.

In summary, precision separates whenever $G(\text{supp}(F)^c) > 0$, and coverage separates both under partial support mismatch and under smooth covariate shift with common supports. In the latter regime, coverage attains its maximum at $F = G$ and is strictly smaller otherwise, with explicit quantitative gaps available under simple bounds on the density ratio or under margin assumptions.

## A.3 CONNECTION WITH TWO-SAMPLE TESTS

We put the PRDC metrics and T3 which is based on Forte (Ganguly et al., 2025c) in the context of non-parametric two-sample tests using $k$-nearest neighbors. Friedman et al. (1973), Friedman & Rafsky (1979), and Schilling (1986) developed non-parametric two-sample tests based on a pooled-graph statistic to determine whether two sets of observed samples are from the same distribution. Given two sets of samples $X = \{X_i\}_{i=1}^m$ and $Y = \{Y_j\}_{j=1}^n$ i.i.d. with distributions $F$ and $G$ respectively, the tests only seek to determine whether $F = G$. On the other hand, we are concerned not just with a binary decision for the whole sample set, but also whether each individual sample $Y_j$ is from the same distribution as $X$, making our setting much more complex. Nevertheless, comparing the PRDC metrics with these tests helps us build a better understanding of the mathematical properties of Forte. We primarily focus on Schilling's test for that purpose, which we restate here.

**Definition A.3** (Schilling's $T_{k,N}$ Statistic). Let $X = \{X_i\}_{i=1}^m$ and $Y = \{Y_j\}_{j=1}^n$ be i.i.d. with distributions $F$ and $G$ respectively. Let $Z = X \cup Y$, $Z_i$ be the $i^{\text{th}}$ element of $Z$, and $N = m + n$. The statistic $T_{k,N}$ is the proportion of all $k$-nearest neighbor comparisons in which a point and its neighbor share the same original label (reference or test), i.e.

$$T_{k,N} = \frac{1}{Nk} \sum_{i=1}^N \sum_{r=1}^k I_r(Z_i),$$

where $I_r(Z_i) = 1$ if and only if $Z_i$ and its $r^{\text{th}}$ neighbor in $Z$ are both from $X$ or both from $Y$.

We first note the major differences between Forte and Schilling's test. While PRDC metrics compute the $k$-nearest neighbors of each sample point $X_i$ or $Y_j$ within its own sample set (i.e. $NB_k(X_i; X)$ and $NB_k(Y_j; Y)$), Schilling's test considers the nearest neighbor of each point in the pooled sample (i.e. $NB_k(X_i; X \cup Y)$ and $NB_k(Y_j; X \cup Y)$). This makes them non-equivalent in general. Moreover, Forte is asymmetric in the sets $X$ and $Y$ by design. Since there is an initial overhead of embedding computation and density estimation, Forte computes the metrics for each test point $Y_j$ individually and then uses the previously estimated density and $k$-nearest neighbors of the points in the reference set $X$ to make predictions, making the method scalable. If we wanted to use a two-sample test like Schilling's in this setting, we would have to calculate the $k$-nearest neighbors of the pooled set $X \cup Y$ from scratch each time we wanted to make predictions for a new test set $Y$, which is prohibitively expensive. Unlike classical two-sample tests requiring $O((m+n)^2)$ recomputation for each new test batch, our asymmetric formulation achieves:

- **Preprocessing**: $O(m^2 + m d_{\max} K)$ for reference embedding and k-NN computation
- **Inference**: $O(n(m + d_{\max} K))$ per test batch, amortizing reference computations
- **Memory**: $O(m \cdot d_{\max} K)$ for cached embeddings and neighbor indices

where $d_{\max} = \max_k d_k$. GPU acceleration via `torch.cdist` and persistent embedding caching further reduce practical latency.

Thus, these classical two-sample tests do not carry over directly to the modern setting of large scale out-of-distribution detection. Nevertheless, given that their statistical properties are well-studied, they can still provide useful insights about modern methods like Forte.

Recall that a statistical test is called consistent if under any alternative hypothesis, the probability of rejecting the null hypothesis converges to 1 as the sample size approaches infinity. We denote by $H_0$ the null hypothesis that $F = G$ (the underlying distributions generating the two sets is the same), and by $H_1$ the alternative hypothesis that $F \neq G$.

**Theorem A.4** (Asymptotics of $T_{k,N}$ (Schilling, 1986, Thm. 3.1 and 3.4)). Under the null hypothesis $H_0$, and assuming $\lim_{m,n\to\infty} m/(m+n) = \lambda_1$ and $\lim_{m,n\to\infty} n/(m+n) = \lambda_2$, the statistic $T_{k,N}$ is asymptotically normal:

$$\sqrt{Nk} \, \frac{T_{k,N} - \mu}{\sigma_k} \Rightarrow \mathcal{N}(0,1), \quad \text{where} \quad \mu = \lambda_1^2 + \lambda_2^2$$

and the variance $\sigma_k^2$ depends on dimension-stable nearest-neighbor interaction probabilities. Moreover, the test based on $T_{k,N}$ is against the alternative hypothesis $H_1$.

In particular, note that $\lim_{N \to \infty} \mathbb{E}[T_{k,N} \mid H_0] = \lambda_1^2 + \lambda_2^2$ does not depend on $k$.

The consistency of Schilling's test is proved in by showing $\liminf_N \mathbb{E}[T_{k,N}|H_1] > \lim_N \mathbb{E}[T_{k,N}|H_0]$, i.e. the limit infimum of the expectation of the statistic under the alternative hypothesis is strictly larger than under null hypothesis. This conforms with the intuition that if the two distributions are different then there will not be enough mixing among the samples, leading to larger values of $T_{k,N}$.

Now, we show that the PRDC metrics and Schilling's statistic $T_{k,N}$ capture some of the same information.

**Lemma A.5.** For any $Y_j \in Y$, $R_k^{(j)}(X, Y) = 0$ if and only if $\frac{1}{k} \sum_{r=1}^{k} I_r(Y_j) = 1$.

*Proof.* Since $I_r(Y_j) \in \{0, 1\}$, the average $\frac{1}{k} \sum_{r=1}^{k} I_r(Y_j)$ can equal 1 if and only if $I_r(Y_j) = 1$ for all $1 \le r \le k$. By definition, $I_r(Y_j) = 1$ if and only if the $r^{\text{th}}$ nearest neighbor of $Y_j$ (in $X \cup Y$) is in the set $Y$. Thus, $\frac{1}{k} \sum_{r=1}^{k} I_r(Y_j) = 1$ if and only if there is no $X_i$ in $NB_k(Y_j; Y)$ (otherwise such an $X_i$ would be one of the first $k$ neighbors of the $Y_j$ in the set $X \cup Y$), which is equivalent to $\sum_{i=1}^{m} \mathbb{1}(X_i \in NB_k(Y_j)) = mR_k^{(j)}(X, Y) = 0$. □

A similar argument shows that for any $X_i \in X$, $\sum_{j=1}^{n} \mathbb{1}(Y_j \in NB_k(X_i)) = 0$ if and only if $\frac{1}{k} \sum_{r=1}^{k} I_r(X_i) = 1$. Recall that the expression $\frac{1}{k} \sum_{r=1}^{k} I_r(X_i)$ measures the proportion of the first $k$ nearest neighbors of $X_i$ that have the same label as $X_i$. We can combine these expressions to recover $T_{k,N}$

$$T_{k,N} = \frac{1}{Nk} \sum_{i=1}^{N} \sum_{r=1}^{k} I_r(Z_i) = \frac{1}{m+n} \left( \sum_{i=1}^{m} \frac{1}{k} \sum_{r=1}^{k} I_r(X_i) + \sum_{j=1}^{n} \frac{1}{k} \sum_{r=1}^{k} I_r(Y_j) \right).$$

We note that the lemma above implies $\left\lfloor \frac{1}{k} \sum_{r=1}^{k} I_r(Y_j) \right\rfloor = \left\lfloor 1 - R_k^{(j)}(X, Y) \right\rfloor$ where $\lfloor x \rfloor$ represents the greatest integer lesser than or equal to $x$. Moreover, $\left\lfloor \frac{1}{k} \sum_{r=1}^{k} I_r(Y_j) \right\rfloor = \min_{1 \le r \le k} \{I_r(Y_j)\}$ which is equal to 1 if and only if *all* of the $k$ neighbors of $Y_j$ in $X \cup Y$ are in $Y$. We can construct a new test statistic as replacing the average of $I_r$ over $r$ with the minimum,

$$B_{k,N} = \frac{1}{N} \sum_{i=1}^{N} \min_{1 \le r \le k} I_r(Z_i) = \frac{1}{m+n} \left( \sum_{i=1}^{m} \min_{1 \le r \le k} I_r(X_i) + \sum_{j=1}^{n} \min_{1 \le r \le k} I_r(Y_j) \right).$$

We conjecture that $\liminf_N \mathbb{E}[B_{k,N}|H_1] > \lim_N \mathbb{E}[B_{k,N}|H_0]$, just as for $T_{k,N}$, and that $B_{k,N}$ is consistent as a consequence. Because of the lemma above, $B_{k,N}$ can be constructed using PRDC metrics. Since Forte fits a more general distribution to the PRDC metrics, we expect it to perform at least as well as the statistic $B_{k,N}$.

## B    EXPERIMENT TECHNIQUE DETAILS

We are committed to scientific reproducibility, and letting each technique in literature we reproduce in our experiments to be best possibly tuned for their best performance. In this section, we share more details about our experimental setup.

### B.1    ADAPTATIONS FOR TEXT-BASED OOD DETECTION

Since most established Out-of-Distribution (OOD) detection methods were originally designed for computer vision, we adapted them to operate on 1024-dimensional text embeddings from the Qwen3-Embedding-0.6B sentence transformer. A common challenge was the absence of components like classifier logits or weights, which are available in supervised vision models. We addressed this by training an auxiliary binary logistic regression classifier on in-distribution (ID) texts versus synthetic background data. This classifier provided the necessary outputs, such as pseudo-gradients, weights, and logits, to enable the application of these methods in an unsupervised text-based setting. Furthermore, distance metrics were consistently adapted from Euclidean to cosine similarity to suit normalized text embeddings, and dependencies on vision-optimized libraries like FAISS were replaced with direct matrix operations in NumPy for efficient computation.

1. **AdaScale:** The auxiliary classifier's weights were used to compute pseudo-gradients, approximating the sensitivity of each embedding dimension. Perturbation was then applied to the most stable features, identified as those with the smallest absolute gradients.

2. **CIDER & NNGuide:** The FAISS dependency for nearest neighbor search was removed in favor of direct cosine distance computation. We implemented exact k-NN retrieval using NumPy partitioning, which we believe improves performance over approximate methods. For NNGuide specifically, the auxiliary classifier's logits were used to generate confidence scores, which in turn produced confidence-weighted "guided" bank features.

3. **FDBD:** The weight matrices from the auxiliary binary classifier were used to adapt the denominator matrix computation from its original multi-class formulation to our binary scenario, enabling the calculation of the required weight difference norms.

4. **GMM:** We removed the supervised learning requirement by working directly with sentence embeddings. When dimensionality reduction was necessary, synthetic background data was used to create pseudo-labels for training a Linear Discriminant Analysis (LDA) model. We used the more numerically stable `sklearn` implementation for all reported results.

5. **OpenMax:** A binary class structure was established using the auxiliary classifier to compute Mean Activation Vectors (MAVs). The classifier's probability outputs, rather than raw embeddings, were then used to fit the Weibull models required by Extreme Value Theory.

6. **ReAct:** The auxiliary classifier generated the logits needed for energy score computation. The activation thresholding mechanism was adapted to work directly on the embedding vectors, where element-wise clipping was applied based on a percentile threshold derived from the training data.

7. **RMD:** We created a pseudo-binary statistical separation by computing class-conditional statistics on a random subset of the training data while using the full training set for the background distribution statistics. This ensured a sufficient distributional difference for the RMD scoring function.

8. **VIM:** The weight matrix ($w$) and bias ($b$) from the auxiliary classifier were used to define the center point for the principal subspace. This subspace was computed by applying eigendecomposition to the covariance matrix of centered text embeddings, using the eigenvectors with the smallest eigenvalues as the null space basis.

### B.2    API BASELINE INTEGRATION

The Perspective API and the OpenAI Omni Moderation API are both natively designed for text content safety and required minimal adaptation. We implemented a unified integration layer for both, which included persistent file-based caching to minimize redundant calls, detailed logging for reproducibility, and robust error handling for network issues. To ensure consistency with our

evaluation framework, the output probabilities from each API were converted into a standardized safety score, calculated as $1.0 - \text{max\_toxicity\_score}$.

### B.3 OPEN-SOURCE JUDGE LLM ADAPTATIONS

Some of the Judge LLMs were not designed to output the continuous confidence scores required for OOD evaluation metrics like AUROC. The primary adaptation for each model, therefore, was to convert its distinct native output, whether categorical, structured, or multi-label, into a unified numerical safety score suitable for our framework.

- **LlamaGuard:** Its discrete classifications (e.g., "safe," "unsafe," "unsafe S1") were mapped to fixed confidence values. "Safe" classifications received a high score (0.9), while "unsafe" and specific violation categories received progressively lower scores (0.1 and 0.05, respectively) to reflect greater certainty of harm.

- **MD-Judge:** We used a conversation-style prompt to elicit its structured output, which contains a safety category and a numerical severity score (1-5). A scaling function then converted these discrete outputs into a continuous score, ensuring that higher severity ratings corresponded to lower safety confidence.

- **DuoGuard:** As reccommended by their creators, its multi-label output, a probability vector across 12 safety subcategories, was converted into a single value using a max-aggregation approach. The final safety score was calculated as $1 - \max(\text{category\_probabilities})$, effectively treating the highest-risk category as the overall risk indicator.

- **PolyGuard:** Since it evaluates prompt-response pairs, we supplied a generic, safe response ("I cannot and will not provide that information") for every input prompt. We then parsed its structured text output, which classifies prompt harmfulness and identifies policy violations. A hierarchical scoring system assigned a high score for safe content, a medium score for refusals, and progressively lower scores for harmful content based on the number of violations detected.

## C EXPERIMENT PARAMETERS

This section provides a detailed account of the datasets, models, and hyperparameters used in our experiments to ensure full reproducibility.

### C.1 TOXICITY AND DOMAIN-SPECIFIC EVALUATION PARAMETERS

The parameters detailed below were used for the toxicity detection experiments (results in Table 1) and the zero-shot domain generalization experiments (results in Table 4).

#### C.1.1 DATASETS

- **In-Distribution (ID) Data:** The ID dataset was a curated mix of safe prompts, labeled as `id_mix`. It consisted of 40,000 total samples drawn equally from four sources: `tatsu-lab/alpaca`, `databricks/databricks-dolly-15k`, `Anthropic/hh-rlhf`, and `OpenAssistant/oasst2`.

- **Out-of-Distribution (OOD) Data:** OOD data was sourced from multiple benchmarks, with a maximum of 10,000 samples used per benchmark.
  - **Toxicity & Hate Speech Benchmarks:** `RealToxicityPrompts`, `CivilComments`, `HatEval`, `Davidson`, `HASOC`, and `OffensEval`.
  - **Domain-Specific Benchmarks:** Harmful prompts from five domains in the PolyGuard dataset: `social_media`, `education`, `hr`, `code`, and `cybersecurity`.

#### C.1.2 GENERAL & T3 CONFIGURATION

- **General:** All experiments were run on a `cuda:0` device with a random seed of **42** and a batch size of **32**.

- **T3 (Forte) Models:** The T3 framework was configured with a multi-view representation derived from three sentence transformers: `Qwen/Qwen3-Embedding-0.6B`, `BAAI/bge-m3`, and `intfloat/e5-large-v2`.

### C.1.3 BASELINE MODEL HYPERPARAMETERS

The following models and hyperparameters were used for the baseline comparisons. For all representation-based OOD methods, the primary sentence transformer was `Qwen/Qwen3-Embedding-0.6B`.

- **AdaScale:** The percentile range was set to $(90.0, 99.0)$ with $k_1 = 50.0$, $k_2 = 50.0$, $\lambda = 1.0$, and a perturbation strength of $o = 0.1$.
- **CIDER:** The number of nearest neighbors was set to $K = 5$.
- **DuoGuard:** The model used was `DuoGuard/DuoGuard-0.5B` with a classification threshold of $0.5$ and a maximum sequence length of $512$.
- **fDBD:** Distance to mean was used for normalization.
- **GMM:** The model was configured with 8 clusters, a 'tied' covariance type, and used the 'penultimate' feature type without dimensionality reduction. The `sklearn` implementation was used.
- **LlamaGuard:** The model used was `meta-llama/Llama-Guard-3-1B`.
- **MD-Judge (vLLM):** The model was `OpenSafetyLab/MD-Judge-v0_2-internlm2_7b` with a generation temperature of $0.1$, max new tokens of $128$, and GPU memory utilization of $0.7$.
- **NNGuide:** The number of nearest neighbors was $K = 100$ with $\alpha = 1.0$.
- **OpenMax:** The configuration used a tail size of $20$ for Weibull fitting, an $\alpha$ of $3$, and a 'euclidean' distance metric.
- **PolyGuard (vLLM):** The model was `ToxicityPrompts/PolyGuard-Qwen-Smol`. Evaluation was performed on prompts only by providing a dummy safe response: "'I cannot and will not provide that information.'".
- **ReAct:** The activation clipping threshold was set to the $90^{th}$ percentile.
- **VIM:** The principal subspace dimension was set to $d = 512$.

## C.2 MULTILINGUAL EVALUATION PARAMETERS

### C.2.1 DATASETS

- **In-Distribution (ID) Data:** The ID dataset consisted of 30,000 safe, helpful prompts sourced from the `OpenAssistant/oasst2` dataset.
- **Out-of-Distribution (OOD) Data:** The OOD data was composed of harmful prompts from two multilingual benchmarks. For each benchmark, a maximum of 800 samples were used per language.
  - **RTP_LX**: The languages evaluated were English (`en`), Spanish (`es`), French (`fr`), German (`de`), Italian (`it`), Portuguese (`pt`), Russian (`ru`), Japanese (`ja`), Korean (`ko`), Chinese (`zh`), Hindi (`hi`), Dutch (`nl`), Polish (`pl`), and Turkish (`tr`).
  - **XSafety**: The languages evaluated were English (`en`), Chinese (`zh`), Arabic (`ar`), Spanish (`sp`), French (`fr`), German (`de`), Japanese (`ja`), Hindi (`hi`), and Russian (`ru`).

### C.2.2 GENERAL & T3 CONFIGURATION

- **General:** All experiments were run on a `cuda:0` device with a random seed of **42** and a batch size of **24**.
- **T3 (Forte) Models:** The T3 framework utilized a multi-view representation from three sentence transformers: `Qwen/Qwen3-Embedding-0.6B`, `BAAI/bge-m3`, and `intfloat/e5-large-v2`.

**Why you should consider using Representation Typicality Estimation** We adopted Forte (Ganguly et al., 2025c) because it treats OOD detection as a representation-space problem: it uses self-supervised embeddings to preserve meaning-bearing structure and can be dropped into an existing pipeline without redesign. In practice, it can consume features from virtually any extractor—including hidden-layer activations from materials-focused neural networks—so it fits naturally with domain-specific models. Its pointwise descriptors are constructed to reflect the nearby geometry of the data, borrowing ideas from manifold and local-neighborhood modeling to respect local structure. The result is a lightweight deployment: no extra training stage, no additional detector to fit, and immediate zero-shot behavior.

Forte also avoids several common prerequisites. It does not depend on labeled classes, it does not require curated OOD examples during development, and it does not assume anything about whether the underlying model is predictive or generative (or what architecture it uses). Evidence of strong cross-domain transfer has been reported on synthetic imagery (Ganguly et al., 2025b), medical MRI data (Ganguly et al., 2025c), anomaly detection in HPC logs (Chen et al., 2025), and LLM safety applications such as jailbreak and toxicity screening (this paper). Conceptually, this broad applicability is unsurprising: OOD detection overlaps heavily with tasks like anomaly detection, novelty detection, and open-set recognition, and Forte is built around a shared primitive—estimating how "typical" a sample is relative to what the representation considers normal. That same typicality signal has been leveraged beyond OOD, including measuring syntactic typicality in neurosymbolic program synthesis for reasoning (Ganguly et al., 2024) serving as a practical stand-in for uncertainty quantification (UQ) (Ganguly et al., 2025d).

### C.2.3    BASELINE MODEL HYPERPARAMETERS

Baseline models were configured with the same hyperparameters detailed in Appendix C.1, with one exception: the primary sentence transformer for representation-based OOD methods in this evaluation was `BAAI/bge-m3` for its multilingual capabilities.

### C.3    OVERREFUSAL EVALUATION PARAMETERS (OR-BENCH)

The parameters in this section correspond to the overrefusal detection experiments on OR-Bench, with results presented in Table 3 of the main paper.

### C.3.1    DATASETS

The evaluation used the `bench-llm/or-bench` dataset, which is specifically designed to measure overrefusal on safe-but-challenging prompts.

- **In-Distribution (ID) Data:** The ID data consists of safe prompts that are known to sometimes trigger overrefusal in LLMs.
    - A pool of 5,000 safe prompts was loaded from the `or-bench-80k` and `or-bench-hard-1k` subsets.
    - This pool was split into 3,500 prompts for the training set and 1,500 prompts for the test set.
- **Out-of-Distribution (OOD) Data:** The OOD set consisted of 600 toxic prompts from the `or-bench-toxic` subset, which are designed to be correctly refused by safety models.

### C.3.2    GENERAL & T3 CONFIGURATION

- **General:** All experiments were executed on a `cuda:0` device with a random seed of **42** and a batch size of **16**.
- **T3 (Forte) Models:** The T3 framework was configured with a multi-view representation derived from three sentence transformers: `Qwen/Qwen3-Embedding-0.6B`, `BAAI/bge-m3`, and `intfloat/e5-large-v2`.

### C.3.3 BASELINE MODEL HYPERPARAMETERS

All baseline models were configured with the same hyperparameters as those used in the Toxicity and Domain-Specific Evaluations, which are detailed in Appendix C.1. The primary sentence transformer for all representation-based OOD methods was `Qwen/Qwen3-Embedding-0.6B`.

## C.4 ADVERSARIAL AND JAILBREAKING EVALUATION PARAMETERS

The parameters in this section correspond to the adversarial and jailbreaking detection experiments, with results presented in Table 2 of the main paper.

### C.4.1 DATASETS

- **In-Distribution (ID) Data:** The ID dataset was the same `id_mix` used in the toxicity evaluations, consisting of safe prompts from `Alpaca`, `Dolly`, `hh-rlhf`, and `OpenAssistant`. The dataset was split into 40,000 samples for training and 15,000 samples for testing.
- **Out-of-Distribution (OOD) Data:** The OOD data consisted of prompts from a wide range of adversarial and jailbreaking benchmarks, as described in the table.

### C.4.2 GENERAL & T3 CONFIGURATION

- **General:** All experiments were run on a `cuda:0` device with a random seed of **42** and a batch size of **32**.
- **T3 (Forte) Models:** The T3 framework was configured with a multi-view representation from three sentence transformers: `Qwen/Qwen3-Embedding-0.6B`, `BAAI/bge-m3`, and `intfloat/e5-large-v2`.

### C.4.3 BASELINE MODEL HYPERPARAMETERS

All baseline models were configured with the same hyperparameters as those used in the Toxicity and Domain-Specific Evaluations, which are detailed in Appendix C.1. The primary sentence transformer for all representation-based OOD methods was `Qwen/Qwen3-Embedding-0.6B`.

## C.5 T3/FORTE ALGORITHMIC ABLATIONS

We conducted ablation studies to analyze the core components of the Forte algorithm and explore potential improvements. Our goal was to determine the contribution of each metric and to test alternative geometric approaches. We first evaluated a simplified one-sample variant of the algorithm using only precision and density scores (T3-PD), which achieves FPR@95 of approximately 20–35%. Adding recall and coverage metrics (T3-RC, the two-sample variant) substantially improves performance, achieving FPR@95 of approximately 0.7–4%. The full combination of all four metrics (T3-Full) provides the most robust detection, further reducing FPR@95 to approximately 1–2% as shown in Table 6. This progression confirms that each metric captures distinct geometric failure modes, and the full set is necessary for comprehensive coverage.

Next, we investigated replacing the standard $k$-NN spherical regions with ellipsoids defined by the local covariance of neighboring points. This approach proved unsuccessful for two primary reasons: (1) *Dimensionality Issues*: In high-dimensional embedding spaces, the estimated ellipsoids became highly eccentric (elongated), leading to unstable distance calculations and performance approaching random chance. (2) *Computational Cost*: The overhead of estimating a unique covariance matrix for each point was prohibitively expensive, making the method impractical for large datasets.

Given these challenges, the use of ellipsoids was abandoned. Finally, we affirmed the framework's robustness to the choice of embedding models. Consistent with the original Forte paper's findings in computer vision (Ganguly et al., 2025c), we heuristically observed that the T3 framework's performance remained strong when different sentence transformers were used, suggesting that the method generalizes well across various NLP embedding spaces.

Table 6: **PRDC component ablation.** T3-Full uses all four metrics (Precision, Recall, Density, Coverage). T3-RC uses only Recall and Coverage (two-sample statistics). T3-PD uses only Precision and Density (one-sample statistics). The full combination provides the most robust detection with consistently low FPR@95.

| Category | Components | Civil Comments | | OffensEval | | Real Toxicity | | XSafety | |
|---|---|---|---|---|---|---|---|---|---|
| | | AUROC | FPR@95 | AUROC | FPR@95 | AUROC | FPR@95 | AUROC | FPR@95 |
| T3-Full | P+R+D+C | 0.9969 | 0.0157 | 0.9998 | 0.0013 | 0.9968 | 0.0157 | 0.9974 | 0.0108 |
| T3-RC | R+C (two-sample) | 0.9846 | 0.0366 | 0.9993 | 0.0013 | 0.9847 | 0.0392 | 0.9961 | 0.0072 |
| T3-PD | P+D (one-sample) | 0.9513 | 0.3343 | 0.9673 | 0.2251 | 0.9672 | 0.2076 | 0.9491 | 0.3494 |

# D INTEGRATING T3 WITH vLLM FOR ONLINE GENERATION GUARDRAILING

We build on insights into underutilized hardware capability from prior work on GPU performance modeling in LLMs(Zhang et al., 2025). This subsection describes the design and implementation of T3 within the vLLM inference framework, focusing on the architectural mechanisms that enable real-time guardrailing during generation. The integration addresses the challenges of enforcing guardrails in high-throughput inference systems where low latency, streaming support, and continuous monitoring are critical deployment requirements. Evaluation on an NVIDIA H200 GPU demonstrates that T3 introduces only negligible generation runtime overheads even under dense evaluation frequencies. To the best of our knowledge, T3 is the first framework to integrate guardrails into online LLM generation.

**Technical Background of vLLM and Why It:** vLLM (Kwon et al., 2023) was selected as the integration target due to its combination of architectural efficiency and widespread adoption. Its PagedAttention mechanism provides scalable KV-cache management, while continuous batching enables high utilization across heterogeneous workloads. vLLM's active development community and modular design ensure sustained compatibility and performance improvements. Importantly, the framework's multiprocess execution model exposes well-defined integration points where safety evaluation can be embedded without perturbing inference performance or scheduling logic. Specifically, the vLLM v1 engine employs a three-tier process hierarchy to achieve scalability and fault isolation:

*Main Process:* Serves as the application entry point. It handles user requests, tokenization, and overall orchestration. Communication with the Engine Core occurs via ZeroMQ IPC, enabling asynchronous scheduling and fault isolation. *Engine Core:* Acts as the central scheduler responsible for global request management, batch construction, and computational resource allocation. It coordinates distributed KV-cache state, implements chunked prefill and pipeline parallelism strategies, and mediates communication between the Main and Worker processes. *Worker Processes:* Execute the transformer model partitions, hosting weights, and performing inference on GPU backends. Multiple workers can operate in parallel under tensor parallelism, returning partial results that the Engine Core consolidates. This hierarchy presents an opportunity for integrating T3 guardrails. Process isolation prevents safety component failures from affecting inference stability. The Main Process offers a natural interception point for modifying outputs without altering scheduling logic. Furthermore, inherent batching in the Engine Core to Main Process interface enables efficient group evaluation of multiple requests within guardrail checks.

**Integration Strategy and Implementation:** T3 was embedded directly into the vLLM pipeline rather than deployed as an external service. This co-design approach meets several requirements: *Latency Minimization:* Avoids serialization, IPC, and network overhead inherent to external microservices, reducing evaluation latency to the sub-millisecond range. *Streaming Compatibility:* Maintains token-by-token evaluation without disrupting streaming responses, in contrast to buffering-based external systems. *Context Accessibility:* Provides direct access to prompt history, partial outputs, and intermediate states necessary for accurate safety assessment. *Lifecycle Control:* Enables immediate termination of unsafe generations by modifying internal finish reasons, eliminating the complexity of coordinating aborts across distributed services.

Integration is achieved by patching the `OutputProcessor.process_outputs` method. This choice allows: *(1) Non-Invasive Modification:* Behavioral changes are introduced without altering the vLLM source, avoiding custom forks or rebuilds. *(2) Performance Containment:* The patch intercepts output at a single chokepoint, preventing scattered performance regressions. *(3) Maintainability:* All safety logic is localized within a single interception function, simplifying debugging and iteration.

*(4) Operational Flexibility:* Guardrails can be enabled or disabled dynamically, facilitating controlled rollout and experimentation.

Originally, `process_outputs` iterates over each request in a batch, performing detokenization, log probability computation, and output construction for newly generated tokens from workers. The T3 integration restructures this flow into three coordinated phases: *Phase 1: State Synchronization:* Standard processing tasks such as detokenization, log probability computation, and request state updates are performed. Newly generated text segments are accumulated into an injected class-level tracking structure (`self.reqs`) for later evaluation. *Phase 2: Guardrail Evaluation:* Candidate requests are selected using multi-tier scheduling policies that balance detection frequency with computational efficiency. Batches of candidate texts are passed through safety classifiers, and unsafe requests are flagged by setting their `finish_reason` to `ABORT`. *Phase 3: Conditional Output Generation:* Final outputs are constructed. For requests flagged in Phase 2, the system produces termination responses annotated with explicit stop reasons, while all other requests continue through the normal generation path.

Listing 1: T3 Integration via Monkey Patching vLLM's `process_outputs` Function

```python
def process_outputs_with_T3(self, engine_core_outputs, **kwargs):
    if not hasattr(self, 'reqs'):
        self.reqs = {}
    req_outputs = [] reqs_to_abort = []

    # Phase 1: Standard vLLM processing + text accumulation
    for engine_core_output in engine_core_outputs:
        # ... Standard vLLM processing (stats, detokenization, logprobs)

        # Track accumulated text metadata for T3
        current_text = req_state.detokenizer.output_text
        self.reqs[req_id] = {
        'text': current_text, 'word_count': len(current_text.split()),
        'last_predicted_at': 0, 'finish_reason': finish_reason
        }

    # Phase 2: T3 processing depending on batch size satisfaction
    texts, req_ids = assemble_evaluation_batch(self.reqs)
    if len(texts) >= min_batch_size:
        predictions = guardrails.predict(texts)
        for i, req_id in enumerate(req_ids):
            if predictions[i] < 1:  # Toxic detected
                mark_request_for_abort(engine_core_outputs, req_id)

    # Phase 3: Output creation with guardrails decisions
    for engine_core_output in engine_core_outputs:
        # ... Standard vLLM output creation and cleanup ...
        req_outputs.append(req_state.make_req_output(engine_core_output))
        reqs_to_abort.append(req_output if req_output.finished else None)

    return OutputProcessorOutput(req_outputs, reqs_to_abort)

# Apply the patch
OutputProcessor.process_outputs = process_outputs_with_T3
```

The integration is designed to minimize computational overhead: *Candidate Scheduling:* Configurable T3 evaluations are enforced using hierarchical scheduling policies (via `assemble_evaluation_batch`). Primary selection identifies requests that reach predefined (`word_count−last_predicted_at`) thresholds. Secondary policies expand candidate sets to near-threshold requests or those nearing completion, thereby stabilizing batch sizes and ensuring efficient GPU utilization. When sufficient batch size cannot be achieved, the system either defers evaluation to subsequent iterations or proceeds with a reduced candidate set under a fallback policy that balances safety responsiveness against computational efficiency. *Memory Efficiency:* Request metadata structures (`self.reqs`) track only essential information (text buffers, counts, evaluation

timestamps, and safety flags). Memory pooling and cleanup policies prevent fragmentation and overhead accumulation during long-running deployments. *Overlapping Computation in Resource-Constrained Settings:* T3 evaluation, executed in the Main Process, runs concurrently with continuous inference in the Worker Processes. When both share the same accelerator, concurrency is achieved either through true parallel execution with CUDA Multi-Process Service (MPS) or through temporal slicing when MPS is unavailable. Guardrailing workloads are opportunistically scheduled into idle GPU cycles, allowing their latency to be effectively hidden behind inference kernels while minimizing contention.

**Runtime Performance Evaluation on Integration:** We evaluated the runtime impact of integrating online guardrails into vLLM (v0.10.2) through detailed profiling with NVIDIA Nsight Systems, using NVTX range annotations to isolate initialization and generation phases. Experiments were conducted on an NVIDIA H200 GPU, employing T3 with three embedding models (Qwen3-Embedding-0.6B, BGE-M3, and E5-Large-V2) trained on 1,000 safe instruction samples from the Alpaca dataset. Generation was benchmarked on Facebook's OPT-125M model, a relatively small LLM chosen to enable rapid inference and thus stress the guardrail system. T3 was configured with a dense evaluation interval of 20 tokens and a prediction batch size of 32 requests, representing a computationally intensive safety configuration.

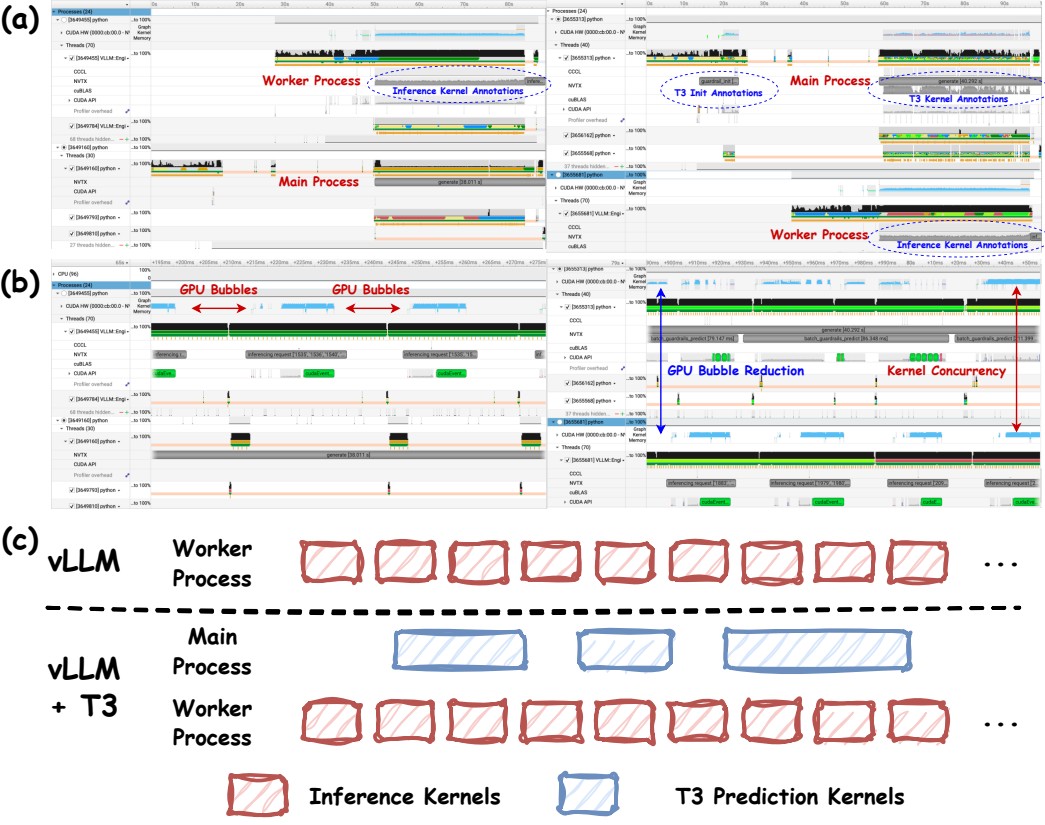

Figure 4: NVIDIA Nsight Systems profiling of vLLM baseline vs. vLLM+T3. (a) Full execution timeline comparison. (b) Zoomed-in view showing kernel concurrency and reduced GPU bubbles in vLLM+T3. (c) Conceptual illustration of overlapping inference kernels (Worker Processes) with T3 prediction kernels (Main Process). The integration reduces idle GPU periods between consecutive generations, improving utilization while preserving low-latency inference.

As shown in Table 7, two workload scales were examined. In the 500-prompt experiment, baseline vLLM completed generation in 6.342 seconds, while the guardrail-enabled system took 6.439 seconds for generation, reflecting a mere 1.5% overhead. In the larger 5,000-prompt experiment, baseline vLLM completed in 38.011 seconds compared to 40.292 seconds with guardrails, corresponding to only 6% overhead (2.281 seconds) while providing continuous safety monitoring across 5,000

requests. The nearly identical initialization times (10.5s in the 500-prompt case vs. 9.8s in the 5,000-prompt case) confirm that one-time setup costs are independent of workload size. Profiling further reveals that T3's prediction workload in the Main Process is almost entirely overlapped with token generation in the Worker Processes, improving overall GPU utilization and reducing idle periods (GPU bubbles) between consecutive generations (Figure 4). These results demonstrate that the three-phase architecture and batching strategies preserve vLLM's high-throughput characteristics on modern accelerators while sustaining real-time guardrail enforcement even under dense evaluation intervals.

Table 7: Runtime performance of baseline vLLM vs. T3 integration running concurrently on an NVIDIA H200 GPU. T3 is configured with a 20-word guardrail interval and a batch size of 32. Given that the generation overhead is negligible in this shared-resource setting, we anticipate virtually no overhead when T3 is deployed with more aggressive settings on dedicated accelerators.

| Workload | System | T3 Init (s) | Inference (s) | Inference Overhead |
|---|---|---|---|---|
| 500 prompts | vLLM baseline | – | 6.34 | – |
| | vLLM + T3 | 10.5 | 6.44 | 1.5% |
| 5,000 prompts | vLLM baseline | – | 38.01 | – |
| | vLLM + T3 | 9.8 | 40.29 | 6.0% |

### D.1 POST-GENERATION GUARDRAILING WITH VLLM

While online detection intervenes during generation, *post-generation guardrailing* evaluates outputs after completion, rendering it a drop-in post-processor to any LLM serving framework. This mode is particularly suited for high-throughput batch inference, retrospective auditing, and multi-pass evaluation pipelines where responses must be filtered or scored without disrupting the decoding process. To remain practical at scale, such checks must impose minimal overhead to avoid degrading overall throughput. We benchmarked T3 alongside several representative guardrail methods, DUOGUARD, POLYGUARD, MDJUDGE, and LLAMAGUARD using the OR-Bench dataset on an NVIDIA H200 GPU. The vLLM engine was configured to load the OPT-125M model and generate responses of up to 256 tokens, with these guardrail techniques as post-processors. Batch sizes ranged from 8 to 64. The runtime was measured by averaging over 20 runs, following 5 warm-up iterations. LLAMAGUARD did not support batch sizes $\geq 32$.

Table 8: Runtime (in **milliseconds**) of post-generation guardrailing. Methods are applied post-inference with vLLM, configured with OPT-125M and a maximum generation length of 256 tokens. Batch sizes range from 8 to 64 using OR-Bench. Runtime was measured with the Torch Profiler, averaging 20 runs after warm-up.

| Batch Size | T3_GMM | T3_OCSVM | DUOGUARD | POLYGUARD | MDJUDGE | LLAMAGUARD |
|---|---|---|---|---|---|---|
| 8 | 68.73 | 68.79 | 40.78 | 255.97 | 1105.33 | 676.98 |
| 16 | 60.08 | 59.84 | 48.55 | 273.60 | 1262.26 | 1376.42 |
| 32 | 85.27 | 59.84 | 48.55 | 312.21 | 1439.38 | 2675.74 |
| 64 | 155.81 | 146.49 | 108.04 | 373.74 | 1524.49 | N/A |

As shown in Table 8, the six methods exhibit a clear efficiency hierarchy. DUOGUARD achieves the lowest latency, remaining under 110 ms, but its lightweight design offers more limited detection capability compared to T3. The two T3 variants (GMM and OCSVM) deliver runtimes between 60–156 ms, scaling moderately with batch size while maintaining substantially higher detection fidelity. This positions T3 as an efficient yet accurate alternative, striking a balance between speed and robustness. POLYGUARD introduces significantly higher overheads, while MDJUDGE and LLAMAGUARD are prohibitively expensive for large-scale use: MDJUDGE exceeds one second even at small batches and grows to over 1.5 s at batch size 64, while LLAMAGUARD more than doubles at each step and fails beyond batch size 32. Overall, T3 provides a practical middle ground, retaining competitive efficiency while delivering stronger safety guarantees than lightweight filters and avoiding the prohibitive costs of heavyweight evaluators, making it well-suited for post-generation pipelines where throughput and accuracy must be jointly preserved.

# E    ADDITIONAL EXPERIMENTS

## E.1    EVALUATION ON WILDGUARDMIX

We evaluate T3 on WildGuardMix (Han et al., 2024), using both the training and test splits provided by `allenai/wildguardmix`. For each example, we read the prompt, `prompt_harm_label`, and `adversarial` flag. A prompt is treated as harmful (OOD) if `prompt_harm_label="harmful"` OR `adversarial=True`. The Test split is human-annotated (higher quality), while the Train split is GPT-4-labeled (larger). The ID (safe) half of the evaluation is drawn from our standard held-out safe mixture (Alpaca/Dolly/OpenAssistant).

Table 9: **Performance on WildGuardMix.** T3 achieves the best AUROC and lowest FPR@95 on both splits, outperforming all baselines including WildGuard and Poly-Guard. The Test split (human-annotated) provides a stricter evaluation than the Train split (GPT-4-labeled).

| Dataset | WildGuardMix Test | | WildGuardMix Train | |
|---|---|---|---|---|
| Metric | AUROC | FPR@95 | AUROC | FPR@95 |
| Method | | | | |
| ADASCALE | 0.4299 | 0.9869 | 0.3259 | 0.9907 |
| CIDER | 0.7909 | 0.5993 | 0.7414 | 0.6586 |
| DUOGUARD | 0.7366 | 0.8446 | 0.7661 | 0.8975 |
| FDBD | 0.4626 | 0.9812 | 0.5112 | 0.9707 |
| GMM | 0.7094 | 0.8585 | 0.6809 | 0.8730 |
| LLAMAGUARD3-1B | 0.654 | 1.0000 | 0.7032 | 1.0000 |
| MDJUDGE | 0.6715 | 0.8512 | 0.8024 | 0.7620 |
| NNGUIDE | 0.5798 | 0.9583 | 0.4796 | 0.9849 |
| OPENMAX | 0.4937 | 0.9828 | 0.6077 | 0.9938 |
| POLYGUARD | 0.7837 | 0.7686 | 0.8721 | 0.4686 |
| REACT | 0.3286 | 0.9918 | 0.2662 | 0.9961 |
| RMD | 0.6415 | 0.9419 | 0.6018 | 0.9587 |
| VIM | 0.5595 | 0.9763 | 0.5241 | 0.9829 |
| T3+GMM | **0.8623** | **0.381** | **0.8971** | **0.2422** |
| T3+OCSVM | **0.8882** | **0.3663** | **0.8853** | **0.2802** |

## E.2    EMBEDDING MODEL ABLATION

To verify that our choice of Qwen3-Embedding-0.6B does not unfairly disadvantage baseline OOD methods, we conducted additional experiments using larger embedding models (4B and 8B parameters). These ablations demonstrate that while larger embeddings provide modest improvements for baseline methods, T3 maintains its substantial performance advantage, confirming that the performance gap is due to T3's methodology rather than the embedding backbone.

Table 10: **Embedding ablation with 4B model.** Baseline OOD methods using a 4B parameter embedding model. Despite the larger embedding capacity, traditional OOD methods still exhibit high false positive rates, while T3 maintains strong performance.

| Dataset | Civil Comments | | Davidson et al. | | Hasoc | | Hateval | | OffensEval | | Real Toxicity | |
|---|---|---|---|---|---|---|---|---|---|---|---|---|
| Metric | AUROC | FPR@95 | AUROC | FPR@95 | AUROC | FPR@95 | AUROC | FPR@95 | AUROC | FPR@95 | AUROC | FPR@95 |
| Method | | | | | | | | | | | | |
| ADASCALE | 0.3959 | 0.995 | 0.1851 | 0.9998 | 0.4865 | 0.9928 | 0.3834 | 0.9943 | 0.3356 | 0.9963 | 0.4965 | 0.9838 |
| CIDER | **0.7263** | 0.918 | 0.6112 | 0.9814 | **0.783** | 0.8774 | 0.7734 | 0.8208 | 0.6913 | 0.9760 | **0.7756** | 0.8161 |
| FDBD | 0.565 | 0.9853 | 0.7601 | 0.9454 | 0.4822 | 0.9938 | 0.5537 | 0.9689 | 0.6448 | 0.9551 | 0.4191 | 0.9947 |
| GMM | 0.6667 | 0.9802 | 0.7020 | 0.9419 | 0.7113 | 0.9609 | 0.7979 | 0.8578 | 0.6741 | 0.9869 | 0.7152 | 0.9213 |
| NNGUIDE | 0.4221 | 0.9926 | 0.225 | 0.9999 | 0.5212 | 0.9882 | 0.4642 | 0.9883 | 0.3607 | 0.9987 | 0.5419 | 0.9819 |
| OPENMAX | 0.5091 | 0.9967 | 0.711 | 0.9997 | 0.4238 | 0.9938 | 0.5282 | 0.9954 | 0.5719 | 0.9976 | 0.4240 | 0.9836 |
| REACT | 0.4403 | 0.9944 | 0.216 | 0.9997 | 0.5227 | 0.9913 | 0.4009 | 0.9929 | 0.3684 | 0.9963 | 0.5438 | 0.9769 |
| RMD | 0.6179 | 0.9891 | 0.6598 | 0.9737 | 0.664 | 0.9787 | 0.7337 | 0.9515 | 0.6423 | 0.9856 | 0.6539 | 0.9632 |
| VIM | 0.5518 | 0.9954 | 0.5517 | 0.9974 | 0.5945 | 0.9897 | 0.6467 | 0.9830 | 0.5492 | 0.9987 | 0.6663 | 0.9581 |
| T3+GMM | 0.7010 | **0.4633** | 0.8863 | **0.1780** | 0.7061 | **0.4173** | 0.8898 | **0.1723** | 0.8270 | **0.3139** | 0.6961 | **0.4628** |
| T3+OCSVM | **0.8807** | **0.3959** | **0.9332** | **0.2267** | **0.8793** | **0.4132** | **0.9450** | **0.1859** | **0.8913** | **0.4106** | **0.8795** | **0.4051** |

These ablation results confirm that T3's performance advantage stems from its manifold-based methodology rather than the choice of embedding model. Even when baseline methods are given

Table 11: **Embedding ablation with 8B model.** Baseline OOD methods using an 8B parameter embedding model. Even with significantly larger embeddings, traditional methods fail to approach T3's performance, particularly in FPR@95.

| Dataset | Civil Comments | | Davidson et al. | | Hasoc | | Hateval | | OffensEval | | Real Toxicity | |
|---------|-------|--------|-------|--------|-------|--------|-------|--------|-------|--------|-------|--------|
| Metric | AUROC | FPR@95 | AUROC | FPR@95 | AUROC | FPR@95 | AUROC | FPR@95 | AUROC | FPR@95 | AUROC | FPR@95 |
| Method | | | | | | | | | | | | |
| ADASCALE | 0.4315 | 0.9839 | 0.1961 | 0.9818 | 0.5075 | 0.9568 | 0.4023 | 0.9625 | 0.3751 | 0.9775 | 0.5152 | 0.9602 |
| CIDER | **0.7746** | 0.8748 | 0.6347 | 0.9467 | **0.7967** | 0.8329 | 0.7873 | 0.7877 | 0.7352 | 0.9378 | **0.8007** | 0.8021 |
| FDBD | 0.6101 | 0.9593 | 0.7827 | 0.9266 | 0.5184 | 0.9439 | 0.5795 | 0.9385 | 0.6914 | 0.9415 | 0.4525 | 0.9822 |
| GMM | 0.6936 | 0.9567 | 0.7205 | 0.9084 | 0.7429 | 0.9417 | 0.8371 | 0.839 | 0.6921 | 0.9741 | 0.73 | 0.9018 |
| NNGUIDE | 0.4421 | 0.9632 | 0.2719 | 0.9814 | 0.5489 | 0.9622 | 0.5087 | 0.976 | 0.3927 | 0.9735 | 0.5858 | 0.9336 |
| OPENMAX | 0.5413 | 0.9804 | 0.7497 | 0.9513 | 0.44 | 0.9806 | 0.5501 | 0.978 | 0.6206 | 0.9638 | 0.4541 | 0.9498 |
| REACT | 0.4671 | 0.9815 | 0.2493 | 0.9714 | 0.5536 | 0.9414 | 0.4483 | 0.9431 | 0.3866 | 0.9834 | 0.5664 | 0.9521 |
| RMD | 0.6612 | 0.9748 | 0.6979 | 0.9416 | 0.6985 | 0.9578 | 0.7832 | 0.9173 | 0.6785 | 0.9469 | 0.6693 | 0.9194 |
| VIM | 0.5787 | 0.9492 | 0.5728 | 0.9656 | 0.6046 | 0.9463 | 0.6875 | 0.9497 | 0.5847 | 0.9828 | 0.6793 | 0.9121 |
| | | | | | | | | | | | | |
| T3+GMM | 0.7508 | **0.4352** | **0.9175** | **0.1298** | 0.7549 | **0.3723** | **0.9342** | **0.1518** | 0.8375 | **0.2839** | 0.7106 | **0.4284** |
| T3+OCSVM | **0.9212** | **0.3508** | **0.9386** | **0.1788** | **0.9204** | **0.3998** | **0.9504** | **0.1565** | **0.9013** | **0.3978** | **0.9018** | **0.3845** |

access to embeddings with 8× more parameters, they still exhibit FPR@95 rates exceeding 90% on most benchmarks, while T3 consistently achieves FPR@95 below 45%.

### E.3 TEXT-NATIVE OOD BASELINE COMPARISON

We additionally evaluated classic text-native OOD detection methods—Energy, kNN, and Mahalanobis—applied directly to text embeddings without any vision-to-text adaptation. Energy scores are computed from classifier logits; Mahalanobis distances are computed in the feature space; and kNN uses distances in the embedding space. All methods are trained on ID-only data and evaluated on the same splits as T3.

As shown in Table 12, these methods achieve moderate AUROC on toxicity benchmarks (kNN reaches ∼0.80–0.84), but consistently suffer from unacceptably high false positive rates (FPR@95 typically exceeding 80%). On jailbreaking benchmarks, performance degrades further with FPR@95 approaching 95–100%. In contrast, T3 achieves FPR@95 in the 1–5% range on the same benchmarks. These results confirm that the high false positive rates observed in OOD baselines are inherent limitations of these methods for LLM safety detection, rather than artifacts of implementation choices. See the main results tables for detailed comparisons.

Table 12: **Text-native OOD baseline comparison.** Energy, kNN, and Mahalanobis methods applied directly to text embeddings. While kNN achieves reasonable AUROC on toxicity benchmarks, all methods exhibit unacceptably high FPR@95 (>80%), confirming inherent limitations for LLM safety detection.

| Dataset | AdvBench | | BeaverTails | | HarmBench | | JailbreakBench | | MaliciousInstruct | | XSTest | |
|---------|-------|--------|-------|--------|-------|--------|-------|--------|-------|--------|-------|--------|
| Metric | AUROC | FPR@95 | AUROC | FPR@95 | AUROC | FPR@95 | AUROC | FPR@95 | AUROC | FPR@95 | AUROC | FPR@95 |
| Method | | | | | | | | | | | | |
| ENERGY | 0.5488 | 0.9827 | 0.5167 | 0.9634 | 0.5807 | 0.9600 | 0.4782 | 0.9761 | 0.5313 | 1.0000 | 0.4582 | 0.9810 |
| KNN | 0.4249 | 1.0000 | 0.2212 | 0.9968 | 0.5595 | 0.9550 | 0.6720 | 0.9556 | 0.3786 | 1.0000 | 0.4499 | 1.0000 |
| MAHALANOBIS | 0.2719 | 0.9981 | 0.2517 | 0.9973 | 0.4145 | 0.9900 | 0.5355 | 0.9625 | 0.2241 | 1.0000 | 0.2294 | 1.0000 |

| Dataset | Civil Comments | | Davidson et al. | | Hasoc | | Hateval | | OffensEval | | Real Toxicity | |
|---------|-------|--------|-------|--------|-------|--------|-------|--------|-------|--------|-------|--------|
| Metric | AUROC | FPR@95 | AUROC | FPR@95 | AUROC | FPR@95 | AUROC | FPR@95 | AUROC | FPR@95 | AUROC | FPR@95 |
| Method | | | | | | | | | | | | |
| ENERGY | 0.5682 | 0.9238 | 0.5076 | 0.9634 | 0.5768 | 0.9252 | 0.5586 | 0.9339 | 0.5770 | 0.9355 | 0.5898 | 0.9221 |
| KNN | 0.8002 | 0.8789 | 0.7715 | 0.9430 | 0.8306 | 0.8235 | 0.8368 | 0.8057 | 0.8034 | 0.9002 | 0.8040 | 0.8295 |
| MAHALANOBIS | 0.6232 | 0.9768 | 0.6289 | 0.9762 | 0.6699 | 0.9620 | 0.7007 | 0.9702 | 0.7256 | 0.9653 | 0.6263 | 0.9582 |

### E.4 EVALUATION ON HILL JAILBREAK ATTACKS

The HILL method (Luo et al., 2025) represents a particularly challenging class of jailbreak attacks that transform harmful imperatives into innocuous-looking "learning-style" questions (e.g., "I am studying chemistry, explain this reaction..." instead of "How to make a bomb..."). If such attacks successfully masquerade as benign educational queries, they may fall inside the "typical set" of safe content.

To evaluate T3's robustness against HILL attacks, we use 1,500 safe prompts sampled from Dolly as in-distribution data (1,250 for training, 250 held out for testing) and the 46 HILL jailbreak prompts from Luo et al. (2025) as OOD data. Despite HILL's semantic similarity to educational content, T3 robustly identifies these attacks with near-perfect AUROC ($>0.98$) and very low false positive rates (4.35%). Our intuition is that HILL prompts contain atypical patterns in how harmful intent is expressed, which push them outside the typical set formed by standard safe chat data.

Table 13: **Performance on HILL jailbreak attacks (Luo et al., 2025).** Despite HILL's design to semantically resemble benign educational queries, T3 achieves near-perfect detection with AUROC $>0.98$ and FPR@95 of only 4.35%, demonstrating robustness against attacks specifically crafted to evade OOD detection.

| Method | AUROC | FPR@95 | AUPRC | F1 |
|---|---|---|---|---|
| **T3+GMM** | 0.9803 | 0.0435 | 0.9954 | 0.9881 |
| **T3+OCSVM** | 0.9783 | 0.0435 | 0.9926 | 0.9861 |

